# The role of auxiliary domains in modulating CHD4 activity suggests mechanistic commonality between enzyme families

Yichen Zhong[1], Hakimeh Moghaddas Sani[1,6], Bishnu P. Paudel[2,3,6], Jason K. K. Low [1], Ana P. G. Silva[1], Stefan Mueller [2,3], Chandrika Deshpande[1], Santosh Panjikar [4,5], Xavier J. Reid[1], Max J. Bedward[1], Antoine M. van Oijen [2,3] ✉ & Joel P. Mackay [1] ✉

CHD4 is an essential, widely conserved ATP-dependent translocase that is also a broad tumour dependency. In common with other SF2-family chromatin remodelling enzymes, it alters chromatin accessibility by repositioning histone octamers. Besides the helicase and adjacent tandem chromodomains and PHD domains, CHD4 features 1000 residues of N- and C-terminal sequence with unknown structure and function. We demonstrate that these regions regulate CHD4 activity through different mechanisms. An N-terminal intrinsically disordered region (IDR) promotes remodelling integrity in a manner that depends on the composition but not sequence of the IDR. The C-terminal region harbours an auto-inhibitory region that contacts the helicase domain. Auto-inhibition is relieved by a previously unrecognized C-terminal SANT-SLIDE domain split by ~150 residues of disordered sequence, most likely by binding of this domain to substrate DNA. Our data shed light on CHD4 regulation and reveal strong mechanistic commonality between CHD family members, as well as with ISWI-family remodellers.

The eukaryotic genome is packed into chromatin through interactions with histone octamers. These octamers comprise two copies each of histones H2A, H2B, H3 and H4, and the DNA-octamer complex – a nucleosome – is the fundamental unit of chromatin. Access to the DNA is necessary for all aspects of genome biology, and this access is provided by a specialised group of ATP-dependent translocases known as chromatin remodellers. These enzymes modulate chromatin structure by sliding, interchanging or evicting histones, thereby regulating the access of other chromatin-associating proteins, including nucleases, transcription factors and polymerases, to DNA in both space and time[1,2].

All chromatin remodellers contain a Super Family 2 (SF2) ATPase domain, and the two lobes of this domain undergo conformational changes during repeated cycles of ATP hydrolysis. Despite this commonality, remodellers are flanked by different *cis* domains, thereby defining several families, including SWI/SNF (mating type switching/sucrose non-fermenting), ISWI (imitation switch), INO80 (inositol), and CHD (chromodomain helicase DNA-binding). Most remodellers share the ability to slide the histone octamer along DNA[3], and recent models suggest that the underlying mechanism of this activity might be shared by all remodellers[4–7].

We previously examined the nucleosome sliding activity of CHD4 (Fig. 1A) and suggested a mechanistic model for this activity based both on our biochemical data[4] and on prior work from other groups on other remodellers[6–8]. In this model, binding of the ATPase

[1]School of Life and Environmental Sciences, University of Sydney, The University of Sydney, NSW 2006, Australia. [2]Molecular Horizons, School of Chemistry and Molecular Bioscience, University of Wollongong, Wollongong, NSW 2522, Australia. [3]Illawarra Health and Medical Research Institute, Wollongong, NSW 2522, Australia. [4]Australian Synchrotron, Clayton, VIC 3168, Australia. [5]Department of Molecular Biology and Biochemistry, Monash University, Clayton, VIC 3800, Australia. [6]These authors contributed equally: Hakimeh Moghaddas Sani, Bishnu P. Paudel. ✉e-mail: vanoijen@uow.edu.au; joel.mackay@sydney.edu.au

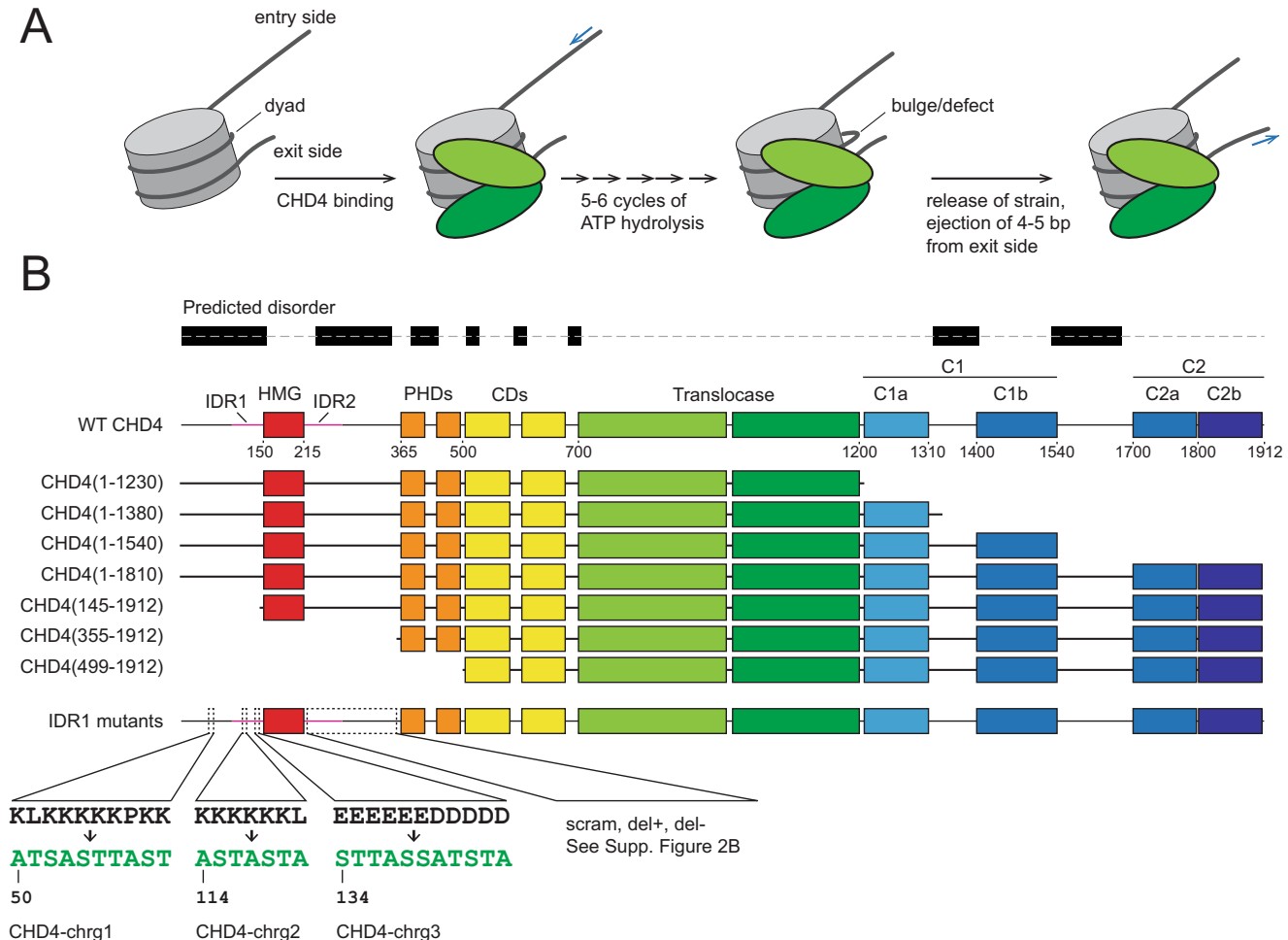

**Fig. 1 | Model for CHD4-driven nucleosome sliding activity. A** Model for CHD4-driven nucleosome sliding, based on previous work[4,19,80]. After initial binding of CHD4, multiple rounds of ATP hydrolysis are coupled to 1-bp translocations of DNA from the entry side, building up a bulge or twist defect that incorporates the additional base pairs. This defect is resolved by the expulsion of DNA from the exit side of the nucleosome. **B** Topology of human CHD4 (Uniprot: Q14839) and truncations used in this study. Regions that are known or predicted to be ordered are shown in colours. Regions predicted by mobiDB (https://mobidb.bio.unipd.it/) to be disordered are indicated as black rectangles. Sequence replacements for IDR1 mutants chrg1, chrg2 and chrg3 are indicated. The chrg4 mutant combines these three mutations together. Sequences for the IDR2 mutants are given in Supplementary Fig. 2B.

domain induces a distortion in one stand of nucleosomal DNA and, through conformational changes in the ATPase lobes driven by ATP hydrolysis, the two strands of extranucleosomal DNA on the so-called entry side are alternately pulled into the nucleosome. After ~4–6 cycles, the strain built up inside the nucleosome is released by the concerted expulsion of a short segment of DNA at the exit side of the nucleosome.

The model that we proposed, however, focuses exclusively on the ATPase/DNA translocase domain. Remodelling activity of this domain alone is severely compromised in vitro[9], suggesting that important elements of the mechanism remain undefined. Figure 1B shows the domain structure of CHD4. The translocase domain is immediately preceded by two plant homeodomains (PHDs) and two chromodomains; together with the ATPase, they form the core region of the remodeller and were the only domains visible in the recent cryoEM structure of full-length CHD4 bound to a nucleosome (though the first PHD domain was not resolved)[10]. The CHD4 PHDs recognize methylated histone H3K9, whereas the chromodomains bind DNA[11] and/or methylated H3K4[12]. An N-terminal HMG-like domain flanked by disordered sequences has been shown to bind nucleic acids, although its role in remodelling is not clear[13]. The C-terminal ~700 residues are predicted to contain two ordered regions separated by ~160 residues that are predicted to be disordered (C1 and C2 in Fig. 1B). Unlike the case for CHD1, no DNA-binding activity has been ascribed to this region, although the C-terminal C2 region (residues 1700−1912; Fig. 1B) is required for connecting CHD with the nucleosome remodelling and deacetylase (NuRD) complex[14].

In this study, we determine the impact of auxiliary domains on the nucleosome sliding activity of CHD4. We demonstrate that an N-terminal intrinsically disordered region (IDR) is essential for full remodelling activity, and that this effect depends more on the amino acid composition than on the sequence order of the IDR. We also combine both bulk and single-molecule data to show that the C-terminal region harbours a regulatory module that comprises both an auto-inhibitory motif and a domain that relieves that inhibition. Together, these data allow us to propose a model for the nucleosome sliding activity of CHD4 that considers all known domains in the protein, highlighting the functional significance of several 'neglected' domains and identifying several previously undefined functional regions. The model also uncovers previously unrecognized similarities with the remodellers CHD1 and ISWI and provides a foundation for future interpretation of disease-causing mutations in CHD4 and its paralogues.

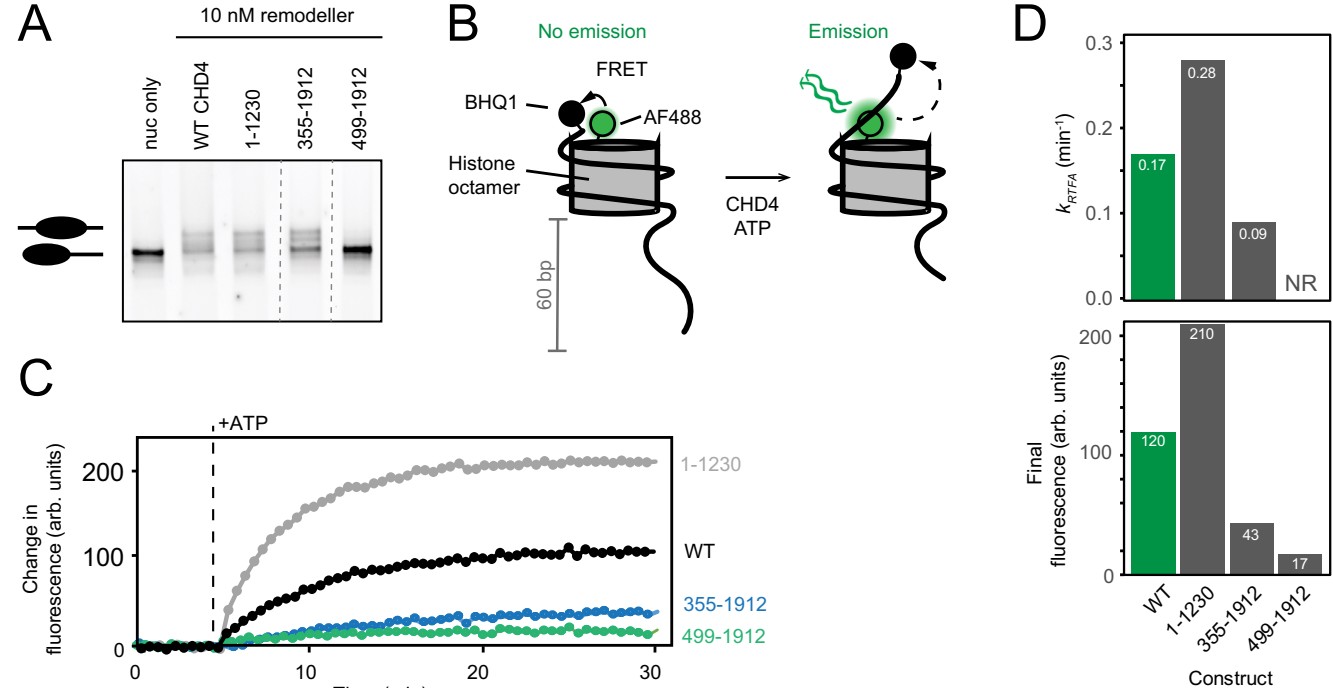

**Fig. 2 | Gel-based and real-time fluorescence assays (RTFA) for CHD4-driven nucleosome sliding. A** Gel based nucleosome sliding assay conducted on WT CHD4 and a range of truncations. In each reaction, a 0w60 nucleosome (60 nM) is treated with 10 nM CHD4 and 1 mM ATP, which induces centring of the histone octamer (creating a 30w30 conformation). The two forms of the nucleosome can be resolved on a polyacrylamide gel, as indicated. Dashed lines indicate where lanes from the same gel have been cut and repositioned. This experiment has been repeated three times; all showed similar results. **B** Set up for real-time fluorescence assay to measure nucleosome sliding. A 0w60 nucleosome is assembled in which H2A is labelled with AF488 (*green*) and the short end of the DNA is labelled with the BHQ1 fluorescence quencher. No fluorescence is observed in this state. Addition of CHD4 ad ATP results in an increase in fluorescence as histone octamer is translocated away from the BHQ1-labelled DNA end. **C** Representative RTFA data for WT CHD4 and several CHD4 truncations. Each trace is an average of at least three technical replicates. CHD4 was mixed with nucleosome before the start of experiment and ATP was added after 5 min (dashed line). **D** Pseudo-first order rate constants and final fluorescence values derived from RTFA data for CHD4 variants (see also Supplementary Fig. S1). Traces were fitted to an asymptotic exponential equation $y = F_{lim}(1-\exp(-k_{RTFA} * t))$, where $F_{lim}$ is the limiting value of the fluorescence, $k_{RTFA}$ is a pseudo-first-order rate constant (units of min$^{-1}$), and $t$ is the time in minutes after addition of ATP. A rate could not be reliably estimated (NR) for CHD4(499–1912). All assays in this and other Figures were repeated at least twice, and representative data are shown. We estimate the uncertainty in fitted parameters to be ~25%. Raw data for **C** are provided as a Source Data file.

## Results

### The N- and C-terminal domains play different roles in CHD4 remodelling activity

We first used a gel-based remodelling assay to assess the roles of auxiliary domains in CHD4. This assay uses a recombinant nucleosome (0w60) containing the Widom positioning sequence with no flanking DNA on one end and a 60-bp extension on the other. Addition of CHD4 shifts the histone octamer along the DNA, converting the asymmetric 0w60 DNA distribution towards a more symmetric 30w30 arrangement; the latter migrates more slowly on native PAGE and at equilibrium, roughly equal concentrations of asymmetric and symmetric conformations are observed[15]. Figure 2A shows that CHD4(1–1230) (which lacks all sequence C-terminal to the DNA translocase domain, Fig. 1B) is highly active as a translocase, whereas CHD4(499–1912) (which lacks the N-terminal HMG-like domain, its flanking intrinsically disordered regions (IDRs) and two PHD domains) exhibited almost no nucleosome sliding activity. An additional construct that retains the PHD domains [CHD4(355–1912)] also displayed reduced activity compared to wild-type CHD4 (WT CHD4).

To better quantify differences in activity, we turned to a real-time fluorescence remodelling assay (RTFA)[16], in which histone H2A in the 0w60 nucleosome is labelled with Alexa Fluor 488 (AF488), while Black Hole Quencher 1 (BHQ1) is conjugated to the shorter end of the nucleosomal DNA (Fig. 2B). Initially, AF488 is in close proximity to the BHQ1, quenching any fluorescence. Following addition of CHD4 and then ATP, histone octamer repositioning increases the BHQ1-AF488 distance and leads to observable fluorescence, which can be measured in real time.

RTFA measurements (Fig. 2C and Supplementary Fig. 1) corroborated the gel-based assays, showing that the activity of the two N-terminal truncations was significantly impaired. Fits to a pseudo-first order model showed that the extent of reaction was reduced by three-fold for CHD4(355–1912) and by seven-fold for CHD4(499–1912) (Fig. 2D). The apparent reaction rate was halved for CHD4(355–1912) and could not be reliably measured for CHD4(499–1912). In sharp contrast and to our surprise, CHD4(1–1230) remodelled the substrate at roughly 1.6 times the rate of WT CHD4 and to twice the extent.

Together, these data demonstrate that the auxiliary domains and disordered sequences flanking the chromodomain-translocase unit of CHD4 impact nucleosome sliding activity of the enzyme and do so in distinct ways.

### The N-terminal intrinsically disordered region regulates remodelling activity

Residues 1–354 of CHD4 harbour a helical HMG-like domain[13], flanked by two regions of >100 residues (IDR1 and IDR2) that are highly charged and predicted to be intrinsically disordered (Fig. 1B). The role of the HMG domain is unknown, although it has been shown that it can bind nucleic acids[13].

We tested the hypothesis that IDR1 and IDR2 might play a role in remodelling. First, three prominent clusters of charged residues in IDR1 were mutated in turn to small/polar residues (Ala, Ser or Thr,

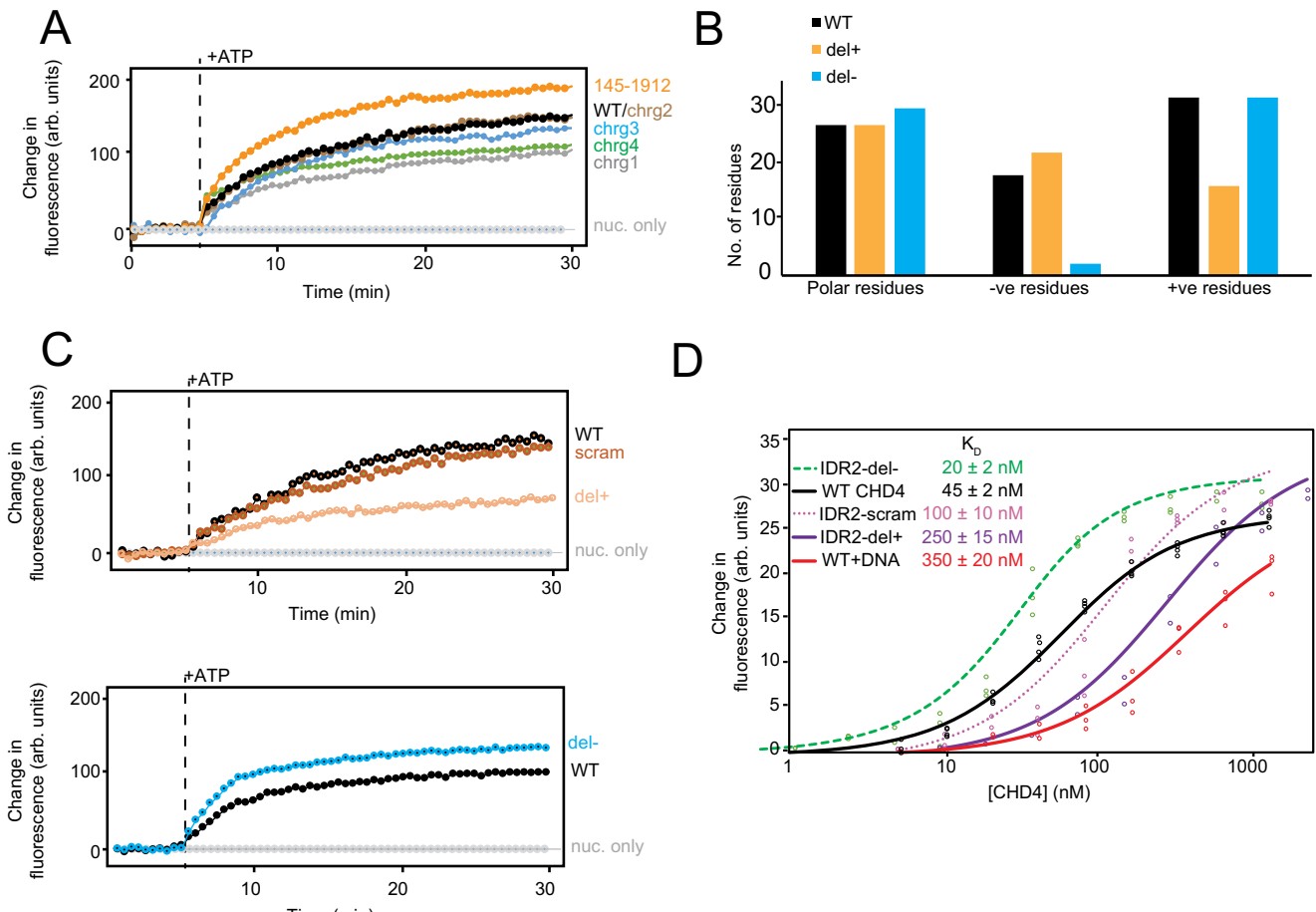

**Fig. 3 | The intrinsically disordered regions (IDRs) in the N-terminal part of CHD4 are important for remodelling. A** RTFA data for the IDR1 charge mutants, along with a mutant in which residues 1–144 were deleted. **B** Numbers of positively or negatively charged residues, as well as polar residues, in the del+ and del– mutants. **C** RTFA data for IDR2 mutants in which the sequence of residues 227–362 was either scrambled (scram) or depleted of either positively charged residues (del + ) or negatively charged residues (del–). We note that the del– curve (which is a mean of three replicates) displays an apparent biphasic shape, but this has not been investigated further at this point. It is only one of the three replicates that has this shape. **D** Microscale thermophoresis (MST) data showing the binding of a 46w60 nucleosome to WT CHD4 and the indicated variants. Three technical replicates and fits to a 1:1 binding isotherm are shown for each mutant and $K_D$ values are indicated (see also Supplementary Fig. S1). We estimate the uncertainty for $K_D$'s to be ~25%. All raw data are provided as a Source Data file.

Fig. 1B and Supplementary Fig. 2A), yielding the constructs CHD4-IDR1-chrg1, -chrg2 and -chrg3; CHD4-IDR1-chrg4 combined all the mutations. In RTFA experiments, CHD4-IDR1-chrg1 and -chrg4 show slightly reduced activity compared to WT, -chrg2 and -chrg3 (Fig. 3A), although all mutants retain considerable remodelling activity (Supplementary Fig. 1). These data suggest that the chrg1 motif, which is rich in lysines, supports nucleosome remodelling activity. Unexpectedly, an N-terminal truncation that completely lacks IDR1, namely CHD4(145–1912), is slightly more active than WT (Fig. 3A). It is thus possible that other elements in the very N-terminal region of CHD4 might inhibit CHD4 activity, but the mechanism underlying this difference is not currently clear.

Because the charged residues in IDR2 are not as clearly clustered, we took a different approach to assessing the involvement of this region in remodelling, creating three distinct mutants. In the first (CHD4-IDR2-scram, Supplementary Fig. 2B), the sequence of IDR2 (residues 227–362) was scrambled randomly so that it had the same residue composition but a different sequence order. In the second (CHD4-IRD2-del + ), we replaced IDR2 with a sequence of the same length from the well-characterized intrinsically disordered protein α-synuclein (Supplementary Fig. 2B). This replacement broadly retains the conformational properties of the region but changes the amino acid sequence and composition – the del+ replacement sequence

harbours similar numbers of polar and negatively charged residues, but only 15 positively charged residues (*cf*. 32 in WT CHD4, Fig. 3B). In the third mutant (CHD4-IRD2-del–), we edited the α-synuclein sequence to reverse the change in charge distribution; the del– replacement sequence eliminates 14 negatively charged residues and included 16 additional positive residues compared to WT but retained the same number of polar residues (Fig. 3B).

Figure 3C shows RTFA data for these three mutants. Strikingly, scrambling of the IDR2 sequence had no significant effect on the rate or extent of nucleosome sliding, whereas the del– substitution yielded a modest increase in remodelling parameters. In sharp contrast, the rate and extent of remodelling were halved for the del+ mutant (Fig. 3C and Supplementary Fig. 1). Together, these data demonstrate that the amino acid composition of IDR2, but not the order of the sequence, is an important contributor to remodelling efficacy – and that it is the density of positively charged residues that dictates that contribution.

To probe the mechanism underlying these effects, we used microscale thermophoresis (MST) to measure the affinity of each CHD4 mutant for a nucleosome. Figure 3D (and Supplementary Fig. S1) shows that the WT CHD4 binds 46w60 nucleosomes with a dissociation constant of 45 nM, in line with our previous estimate from gel-shift assays[4], and significantly stronger than the affinity of CHD4 for naked 46w60 DNA ($K_D = 350 \pm 80$ nM). Scrambling the IDR2 sequence causes

a two-fold decrease in affinity ($K_D = 100 \pm 25$ nM), whereas reducing the number of negatively charged residues (IDR2-del−) increases the affinity by a similar amount ($K_D = 20 \pm 2$ nM). On the other hand, reducing the number of positively charged residues in del+ causes a five-fold decrease in affinity compared to WT CHD4. Together, these data point to a role for the basic residues in both IDR1 (in chrg1) and IDR2 and suggest that IDR2 functions as more than a linker between the HMG and PHD domains. Instead, these basic residues (particularly in IDR2) are an active participant in the remodelling process. Our data suggest that IDR2 achieves this outcome, at least in part, by stabilizing the initial CHD4-nucleosome complex.

## The C-terminal region of CHD4 harbours a substrate binding domain and an auto-inhibitory motif that binds the translocase domain

Almost nothing is known about the region of CHD4 C-terminal to the DNA translocase domain. Like the N-terminal 440 residues, residues 1200–1912 are not observed in the cryoEM structure of the CHD4-nucleosome complex (PDB: 6RYR[10]). It has been suggested that this region does not associate with chromatin[14], thus differentiating it from the DNA-binding domain (DBD) of its paralogue CHD1; it has instead been referred as a Domain of Unknown Function (DUF) in several studies[17–19].

Following our unexpected observation that deletion of residues 1231–1912 increases CHD4 activity in nucleosome sliding assays, we examined a series of C-terminal CHD4 truncations (indicated in Fig. 1B). Deletion of the C-terminal 102 residues (which we have labelled C2b), a region that is predicted to contain two α-helices and to be important for interaction of CHD4 with the rest of the NuRD complex[20], had no effect on the rate or extent of remodelling (Fig. 4A), even though the affinity of CHD4(1–1810) for its substrate was reduced three-fold (Fig. 4B, Supplementary Fig. 1). Deletion of the entire C2 region [CHD4(1–1540)] likewise further reduced nucleosome binding activity (four-fold vs WT CHD4) but did not affect remodelling behaviour. However, a deletion that also removes both C2 and C1b [CHD4(1–1380)] strongly reduced both substrate binding (ten-fold) and remodelling activity (Fig. 4A, B, Supplementary Fig. 1). Deletion to give CHD4(1–1230) restores binding by two-fold – meaning that the affinity of this mutant for its substrate is still five times lower than the wild-type protein and yet the mutant has higher remodelling activity.

The significant difference between the low activity of CHD4(1–1380) and the hyper-activity of CHD4(1–1230) indicates (i) that the C1a region (residues 1231–1380) acts as a strong auto-inhibitory motif and (ii) that the C1b region can release that inhibition to deliver wild-type levels of remodelling in our assay. The stepwise increase in substrate affinity as the C1b, C2a and C2b regions are added suggests that the whole 1381–1912 sequence acts to engage the nucleosomal substrate. The addition of this C-terminal region also restores nucleosome remodelling activity, suggesting that the engagement of DNA by this part of CHD4 might be the event that relieves the C1a-imposed auto-inhibition.

To further test this hypothesis, we examined the interactions between C1a and the translocase domain in WT CHD4. Because C1a is not resolved in the CHD4-nucleosome structure[10], we used covalent crosslinking combined with mass spectrometry (XLMS) to probe the location of C1a in CHD4. We subjected full-length CHD4 to chemical crosslinking using BS3, which yielded a total of 579 high-confidence intra-protein crosslinks (Supplementary Fig. 3A and Supplementary Data 1). Figure 4C depicts the 127 crosslinks (XLs) that involve the autoinhibitory C1a region (residues 1201–1340). Of these XLs, 90 are to residues that lie in the chromodomains or translocase domain, suggesting that C1a might spend a significant portion of time in the vicinity of these domains (Fig. 4C).

Although C1a is absent from the structure of the CHD4-nucleosome complex, and the corresponding region (residues 860–922, termed the bridge) is absent from the structure of the CHD1-nucleosome complex (PDB 5O9G[5], Supplementary Fig. 3B, *left*), the CHD1 bridge *has* been observed in a structure of the CHD1 translocase domain alone (PDB 3MWY[21], Supplementary Fig. 3B, *right*). In this structure, the two lobes of the ATPase domain are in an open arrangement compared to the nucleosome-bound state and residues 860–922 take up a meandering conformation that contacts both lobes of the translocase domain – including residues that, in the CHD1-nucleosome complex, directly contact DNA (Supplementary Fig. 3B, *red*).

Based on this information, we constructed a homology model of the CHD4 chromodomain-ATPase-C1a region in an 'open' conformation, using 3MWY as a template, and mapped the XLs involving C1a onto the model (Fig. 4D, *top*). Although only 50% of the XLs were between residues that were an acceptable distance apart (<35 Å), many of the violated XLs involved regions of C1a that (i) are not closely homologous to CHD1, (ii) are predicted to be disordered and/or (iii) contact a region of the CHD4 translocase domain (such as residues 675–720) that is absent in the cryoEM structure—and therefore likely flexible.

To obtain a more realistic view of the agreement between the XLMS data and the likely solution conformation of CHD4, we used the real-time molecular dynamics (MD) application ISOLDE[22]. ISOLDE allows a user to carry out interactive MD simulations guided by experimental data such as X-ray and cryoEM electron density/potential maps or XLMS data. Portions of the structure can also be restrained to limit their motion during the simulations.

We carried out multiple simulations on our homology model in which the overall structure of the chromodomains and ATPase domain was restrained, with the exception of the 675–725 loop. The conformation of C1a was free to relax under the influence of the XLMS restraints, with the secondary structure of several short segments restrained to be α-helical, in line with predictions from AlphaFold[23], secondary structure predictions for CHD4 and homology with CHD1. Figure 4D (*bottom*) shows the results of these simulations. The C1a region becomes substantially more compact compared to the starting conformation and the 675–725 loop packs into a cleft located between the N- and C-lobes of the ATPase domain. All simulations resulted in similar overall conformations. In these low-resolution models, ~85% of the XLs were satisfied and the location of C1a would clearly inhibit the interaction of the ATPase with DNA in a similar manner to that observed for CHD1.

Together, our remodelling and XLMS data implicate C1a as an auto-inhibitory motif in CHD4 that reduces substrate binding affinity and remodelling activity by binding to and possibly restricting conformational changes in the translocase domain.

## The C1b and C2a regions form a bipartite SANT-SLIDE domain module that can bind double-stranded DNA

Our data show that the region of CHD4 that is C-terminal to C1a (*i.e.*, residues 1340–1912) promotes remodelling activity and contributes an order of magnitude to the substrate binding capacity of CHD4. To determine the mechanism underlying this activity, we sought the structure of this region. After generating many constructs covering the region 1200–1912, we were able to successfully express and purify a protein comprising residues 1380–1810, but lacking residues 1552–1689 [Supplementary Fig. 4[24]]. These residues form part of what is predicted by PONDR[25] to be a disordered segment of ~160 residues in length (residues 1540–1700). We determined the three-dimensional structure of this polypeptide by X-ray crystallography to a resolution of 2.9 Å (Fig. 5A and Supplementary Table 1). The structure strongly resembles the C-terminal DNA-binding domain of CHD1 that contains SANT and SLIDE domains[26,27], although there are topological differences. First, no clear electron density observed for the residues that constitute the bulk of the second helix of the SANT domain (α2 in human CHD1, Fig. 5A). Second, a β-hairpin that links α3 and α4 in

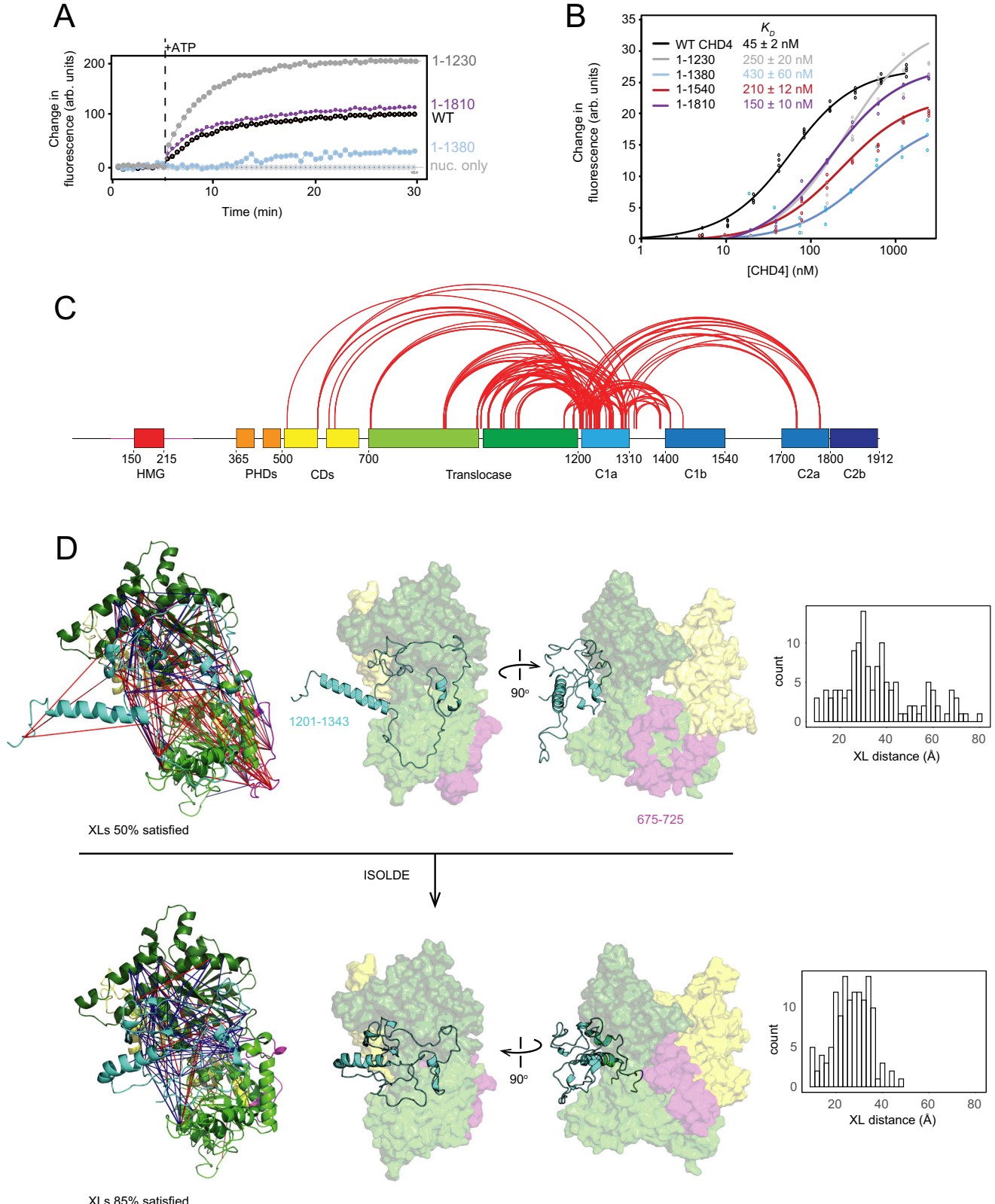

human CHD1 has 'relocated' to a position N-terminal to the first helix in hCHD4; Fig. 5B shows the high sequence similarity between these two elements. In this same region, just N-terminal to α3 of hCHD1, the two proteins do share a ~20-residue loop dominated by charged/small/polar residues—and in both structures this region lacks interpretable electron density, suggesting dynamic behaviour. Yeast CHD1 features a ~70-residue helical insert in this same region (Fig. 5A). Third, the SLIDE

domain features an additional N-terminal helix in hCHD4. The most obvious difference, however, is the absence of electron density for the extended linker sequence (which is 60 residues in length in the truncated construct used for crystallization). A DNA-binding SANT-SLIDE arrangement is also observed in the remodeller ISWI (Fig. 5A, *right*)[28]. In this case, a longer helix connects the SANT and SLIDE elements, and an additional N-terminal HAND domain is present.

**Fig. 4 | The C-terminal third of CHD4 contains an auto-inhibitory domain that binds the DNA translocase. A** RTFA data for C-terminal truncations of CHD4. Data for CHD4(1–1230) are reproduced from Fig. 2C for the purpose of comparison. **B** MST data for C-terminal truncations of CHD4. Data for WT CHD4 are reproduced from Fig. 3E for the purpose of comparison. **C** Covalent crosslinks (XLs) observed for CHD4 in which at least one of the crosslinked residues lies in the C1a region. XLs were visualized using XiNET (https://crosslinkviewer.org/). **D** Data-driven molecular dynamics modelling of the CHD4 XLs involving the C1a region. *Top*. An initial homology model of the CHD4 ATPase domain was created using CHD1 as a template (PDB 3MWY). A combination of the same structure and the AlphaFold

prediction for CHD4 was used to create a starting model the C1a region. Approximately 50% of the XLs were satisfied in this model (*blue* XLs; *red* XLs were violated). *Bottom*. In the interactive molecular dynamics program ISOLDE, manual restraints were used to fix most of the translocase structure and the model was further constrained by the XLs. ISOLDE was then used to search for conformations that better satisfied the XLs than the starting conformation. Multiple models were obtained (only one is shown) that satisfied ~85% of the XLs. The distributions of XL distances (Cα-Cα) in the initial and final models are shown on the *right*. Raw data for **A** and **B** are provided as a Source Data file, XLMS data can be found in ProteomeXchange using identifier PXD033633.

---

The lower panel in Fig. 5A shows electrostatic surface views of the four SANT-SLIDE domains; all three have a prominent basic surface flanked by acidic regions. Both structural and biochemical data on yCHD1 show that this domain binds DNA[26]. As noted above, our nucleosome binding data also point to a role for the C-terminal third of CHD4 in substrate recognition. To test this hypothesis, we titrated 46w60 nucleosomal DNA (without histones) with purified CHD4(1380–1810) in an MST experiment. Figure 5C shows that WT C1bC2a can bind DNA with micromolar affinity, consistent with our hypothesis that the C-terminal region of CHD4 harbours a SANT-SLIDE like DNA-binding domain. Given the role of this portion of CHD4 in relieving C1a-induced autoinhibition, we propose that the SANT-SLIDE domain acts to sense the presence of a nucleosomal substrate. Binding of nucleosomal or extranucleosomal DNA could allosterically disrupt the C1a-ATPase interaction if the geometry of the protein chain cannot fulfil both interactions simultaneously. A similar model has been proposed previously for the ISWI remodeller[29].

### Single-molecule FRET data point to a remodelling mechanism that features two distinct and kinetically observable steps

To further probe the mechanism of CHD4-driven remodelling, we used the single-molecule Förster resonance energy transfer (smFRET) assay that we previously established[4]. A 0w60 nucleosome bearing the FRET pair AF555 (at the '0' end of the nucleosome) and AF647 (on histone H2A) is immobilized on a coverslip and the FRET signal observed by total internal reflection fluorescence (TIRF) microscopy as a function of time (Fig. 6A). Figure 6B shows traces for WT CHD4 at two ATP concentrations. High FRET is observed initially because of the proximity of the fluorophores, and treatment with CHD4 and ATP yields two characteristic sharp drops in FRET that (based on our published FRET calibration data showing that a change of 0.05 in FRET corresponds to a 1-bp shift of the DNA at the exit side[4]) we interpret as concerted ejections of ~4–6 bp of DNA from the exit side of the nucleosome.

In our earlier smFRET study on WT CHD4[4], we examined the distribution of times spent by individual nucleosomes at an intermediate FRET state (which we termed the pause time, $\tau_p$, Fig. 6B). We proposed that during this time multiple rounds of ATP hydrolysis by the DNA translocase domain act to build up an unstable intermediate state in which several base pairs brought into the nucleosome from the entry side are somehow 'stored' (Fig. 1A). Consistent with this idea, a histogram of $\tau_p$ values for WT CHD4 from the current study displays a rise and decay pattern (Fig. 6C). The appearance of such a shape rather than a single exponential signifies that, under those conditions, the remodelling reaction does not have a single rate-determining step (RDS), but rather a set of RDSs with identical (or at least very similar) rate constants[30]. The distribution of pause times then becomes a convolution of multiple single-exponential decays, a distribution that is mathematically described with a gamma function and that shows an initial rise followed by a decay. In our model, the RDSs underlying the rise and decay correspond to the rate constants for successive rounds of ATP-driven 1-bp shifts of the DNA.

A gamma function fit of data for WT CHD4 at 10 μM ATP (Fig. 6C) yields estimates for (i) the number of identical RDSs ($N_{app}$ = 6 to the nearest whole number, in agreement with our previous

measurements[4]) and (ii) the pseudo first order rate constant for each of those steps ($k_N = 0.1\ s^{-1}$). Considering the model outlined in Fig. 1A, this fit suggests that six cycles of ATP hydrolysis, each with a rate constant of $0.1\ s^{-1}$, take place during the pause time. An increase in ATP concentration to 1 mM increases the rate constant three-fold but surprisingly also decreases the apparent number of identical RDSs ($N_{app}$) from 6 to 3. A fit of previous data recorded at a higher ATP concentration (5 mM; from[4]) yields a further increase in $k_N$ to $0.5\ s^{-1}$ and a drop in $N_{app}$ to 1.5.

We previously showed in the general case that, for a process of this type, a reduction in $N_{app}$ is consistent with the 'unmasking' of an additional existing reaction step with a rate constant ($k_X$) that lies within 1–2 orders of magnitude of $k_N$[30]. This additional step could take place either prior to or following the multiple cycles of ATP hydrolysis (Fig. 1A), though it is unlikely to involve binding of CHD4 itself to the substrate, given that the pause time measurements correspond to a state in which the remodeller is already nucleosome bound (**Materials and Methods**). When $k_N \ll k_X$, the $k_X$ reaction step is kinetically 'invisible', the pause-time histogram will have a gamma distribution shape, $N_{app}$ will equal the number of identical RDSs, and the apparent rate constant $k_{app}$ will equal $k_N$ (Supplementary Fig. 5). We propose that this scenario exists at low ATP concentrations. An increase in ATP concentration will raise $k_N$ relative to $k_X$. In the limiting case ($k_N \gg k_X$), $k_X$ will become rate determining, $N_{app}$ will equal 1 and the pause time distribution will fit a single exponential.

Thus, our data suggest a model in which cycles of ATP hydrolysis and DNA translocation (without exit-side DNA expulsion) are rate limiting when ATP level is low, but that a distinct and pre-existing step in the remodelling process becomes rate limiting at high ATP concentration.

### Accessory domains regulate the kinetics of nucleosome remodelling by CHD4

We next tested CHD4(1–1230) and CHD4(1–1540) in the smFRET assay at two different ATP concentrations, monitoring changes in 50–100 individual nucleosomes. In all cases, we observed periods of little or no change in FRET interspersed with sharp decreases, broadly similar to WT CHD4. For both truncations, but particularly CHD4(1–1230), a significant proportion of single-molecule reactions exhibited no measurable intermediate FRET state (Fig. 7A). This effect was more prevalent at the higher ATP concentration, consistent with a lower barrier to the remodelling reaction under those conditions.

In contrast to WT CHD4, the pause time distribution for CHD4(1–1230) fitted to a single exponential at both 10 μM and 1 mM ATP (Fig. 7B, C). This suggests that a single RDS governs the activity of this mutant, consistent with our inference that an additional step exists in the remodelling mechanism that can become rate limiting under conditions where the rate of 1-bp translocations ($k_N$) is increased significantly. In the case of CHD4(1–1230), deletion of the auto-inhibitory domain C1a increases $k_N$ (whereas for WT CHD4 above, the increase in ATP concentration has a similar effect).

CHD4(1–1540), which includes the C1a region but also the SANT portion of the SANT-SLIDE domain, displays behaviour intermediate

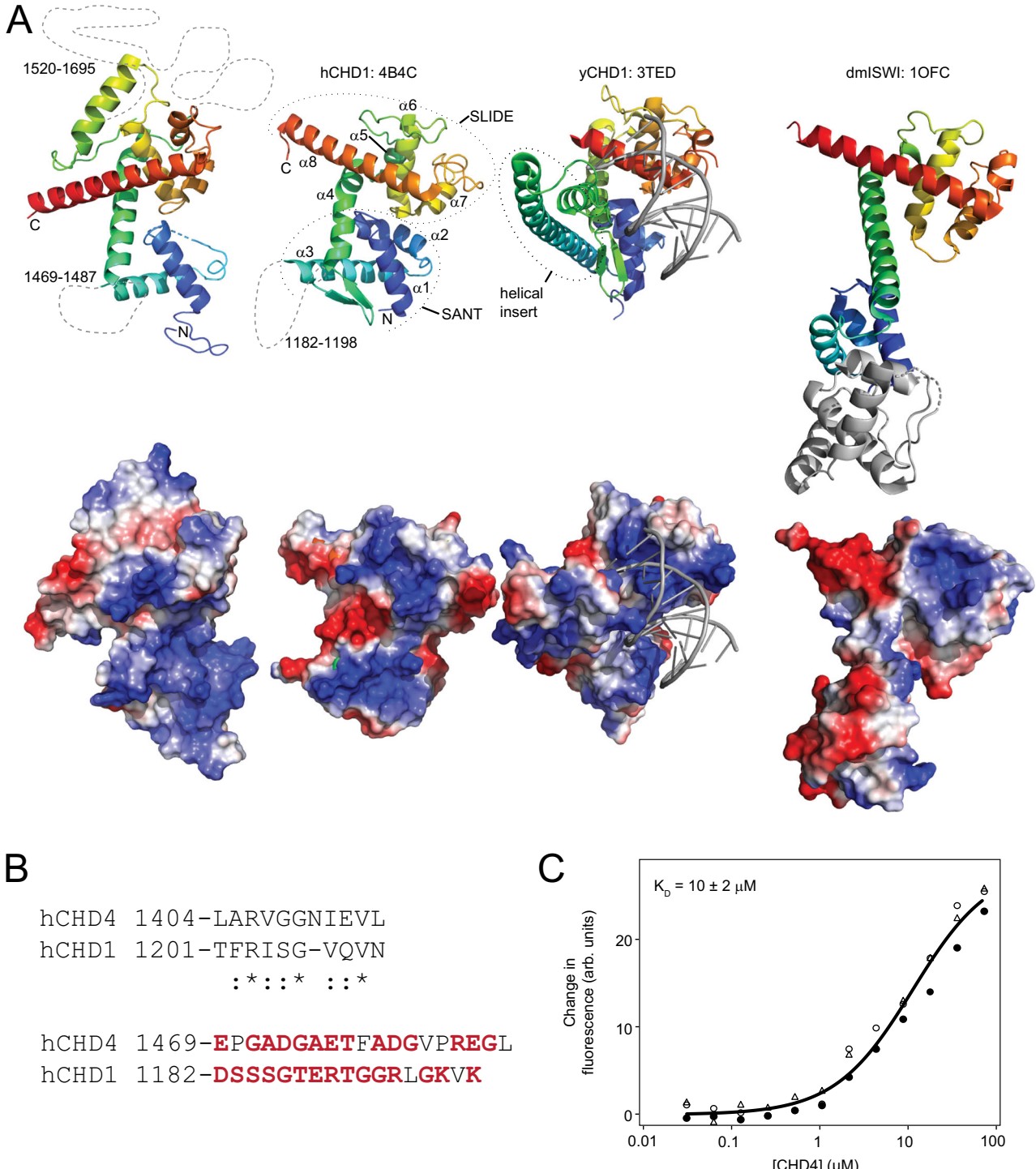

**Fig. 5 | The C1aC2b region contains a SANT-SLIDE domain that can bind DNA.**
**A** *Left*. X-ray crystal structure of C1bC2a-Δ139 (PDB 8D4Y). X-ray crystal structures of the SANT-SLIDE domains of human CHD1 (PDB: 4B4C), yeast CHD1 bound to DNA (PDB 3TED[27]), and the HAND-SANT-SLIDE domain of *Drosophila melanogaster* ISWI (PDB 1OFC[28]) are also shown. In all cases, the ribbon diagram is coloured as a rainbow from N- to C-terminus (*blue* to *red*) and positions and loops for which no electron density was observed are indicated as dashed lines (approximately to scale). Electrostatic surfaces (computed in PYMOL) are shown below each structure. The HAND domain of dmISWI is shown in *grey*. **B** *Top*. Sequence alignment of the β-hairpin region found in the SANT-SLIDE domain of hCHD1 and hCHD4. A high degree of conservation is observed, even though the two motifs are in different positions in the domain topology. *Bottom*. Sequence alignment of the region of the SANT domain that is undefined in both hCHD1 and hCHD4. Both sequences are rich in polar and charged residues (*red*). **C** MST data showing the titration of C1bC2a [CHD4(1380–1800)] into 46w60 DNA. Three technical replicates are shown and the fit to a simple 1:1 binding isotherm is shown as a solid line. Raw data for MST are provided as a Source Data file.

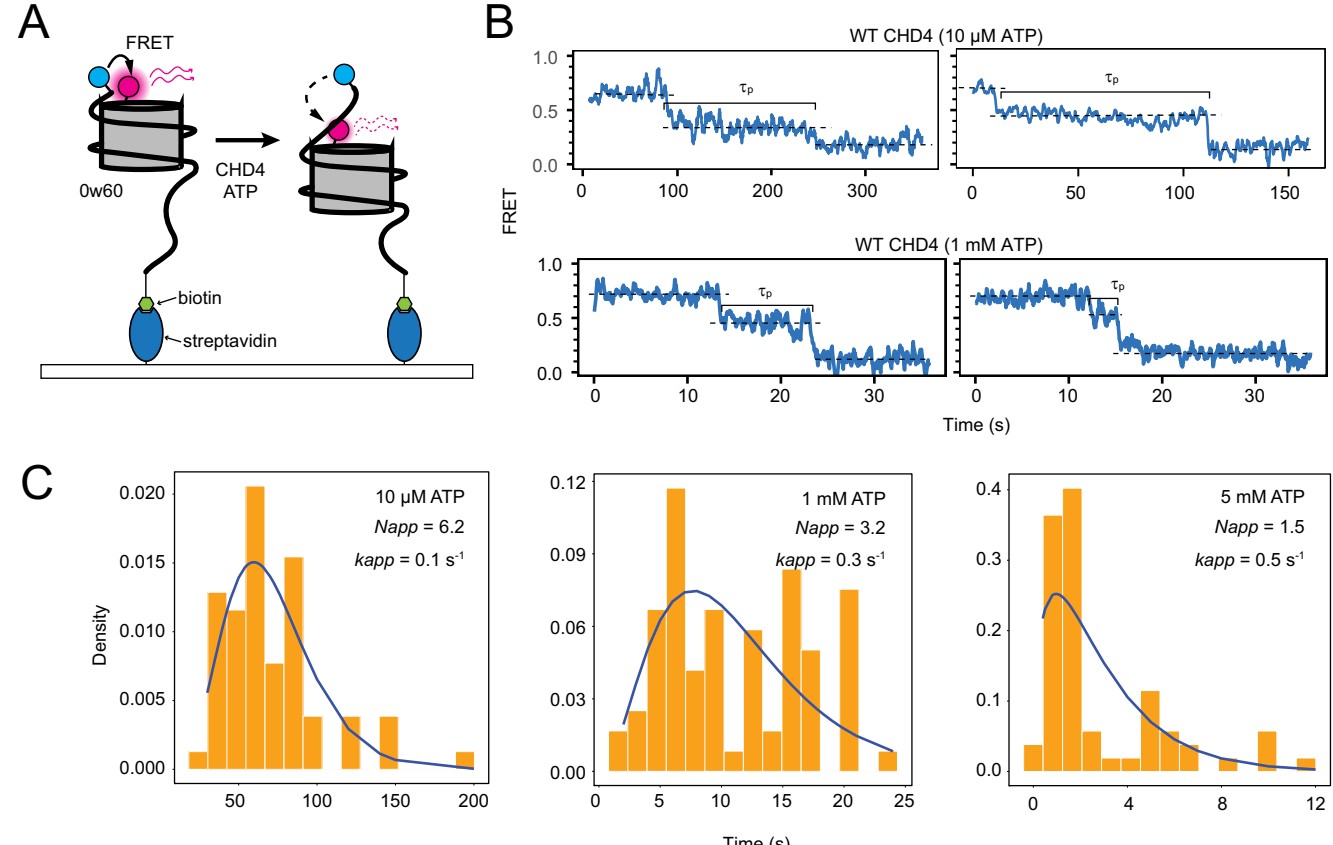

**Fig. 6 | Single-molecule FRET data for WT CHD4. A** Experimental setup for smFRET experiments. For each labelled 0w60 nucleosome, the shorter DNA end is labelled with AF555 (*cyan*) whereas histone H2A is tagged with AF647 (*magenta*) at the T121C position. The nucleosome is fixed on a streptavidin-covered slide uing a biotin tag at the longer DNA end. **B** Typical smFRET traces for WT CHD4 at 10 μM ATP and 1 mM ATP. **C** Histograms of the pause time ($\tau_p$) in the mid-FRET state and fits in each case to a gamma function in which the number of equivalent steps and the pseudo-first-order rate constant are fitted parameters. For all measurements, at least 50 traces were included. All source data for **A**–**C** are provided as a Source Data file.

between the WT and CHD4(1–1230) proteins. At 10 μM ATP, the data are best fit by a gamma function. However, increasing ATP concentration to 1 mM leads to an exponential $\tau_p$ distribution, suggesting that the ratio of $k_N$ to $k_X$ is in this mutant is intermediate between WT CHD4 and CHD4(1–1230) and that, at 1 mM ATP, $k_X$ is rate-limiting (Fig. 7B).

Pause times for the N-terminal truncation CHD4(355–1912) at 1 mM ATP follow a gamma distribution with $N_{app} = 3$, similar to WT CHD4 under the same conditions. The value of $k_{app}$, however, is ten-fold lower for the truncation mutant, reflecting the lower rate observed in our RTFA experiments. Pause time distributions could not be plotted for CHD4(355–1912) at 10 μM ATP or for CHD4(499–1912) because of the small fraction of molecules that proceeded to the lowest FRET state.

For some truncations, such as CHD4(355–1912) and CHD4(499–1912), we also observed that up to 30% of traces showed significant 'reversals' in FRET (examples are shown in Fig. 7A). Although the precise nature of these events is not clear, they likely represent an equilibrium between remodelled and starting conformations of the nucleosome. The latter state is the most thermodynamically stable conformation, given our use of the Widom positioning sequence, and it is therefore possible that the remodelling 'strength' of some mutants is insufficient to counteract the thermodynamic preferences of the nucleosome.

Finally, it is notable that for CHD4(1–1230) the value of $k_{app}$ increases five-fold as the ATP concentration is raised from 10 μM to 1 mM, even though the value of $N_{app}$ remains equal to one. Given that $N_{app}$ should only equal 1 when $k_X$ is rate-limiting ($k_N \gg k_X$), this

observation suggests that the step represented by $k_X$ also depends on ATP concentration. For example, $k_X$ might represent an initial ATP binding event that induces a conformational rearrangement of nucleosome-bound CHD4 to a remodelling-competent conformation. Alternatively, it might represent a 'final' ATP binding event that is associated with the concerted release of DNA from the exit side. However, due to the slow observed rate at 10 μM, we cannot exclude the possibility that additional rate limiting steps are obscured by statistical errors that arise from fluorophore bleaching. Additional mechanistic analysis will be required to address this question.

## Discussion

### Multiple elements of the CHD4 N-terminal domain combine to positively regulate nucleosome sliding

Although the roles of the CHD4 PHD and chromodomains have been examined in detail[31,32], much less is known about the N-terminal 355 residues that comprise two IDRs flanking an HMG-like domain. The HMG domains found in HMGB- and SOX-family proteins bind double-stranded DNA and induce significant DNA bending[33,34]. For SOX proteins, this bending has also been shown to occur in the context of an intact nucleosome, in which case the SOX-bound DNA is lifted away from the histone-DNA interface by ~3–4 Å[35,36], and additional dynamics are induced into the terminal portion of the DNA. Thus, the binding energy provided by the HMG-DNA interaction is sufficient to compensate for the loss of DNA-histone interactions. HMGB proteins contain two HMG domains together with IDRs that are rich in basic or acidic residues and contribute significantly to high-affinity DNA binding[37,38]. It has been hypothesized that these proteins, which are ubiquitously

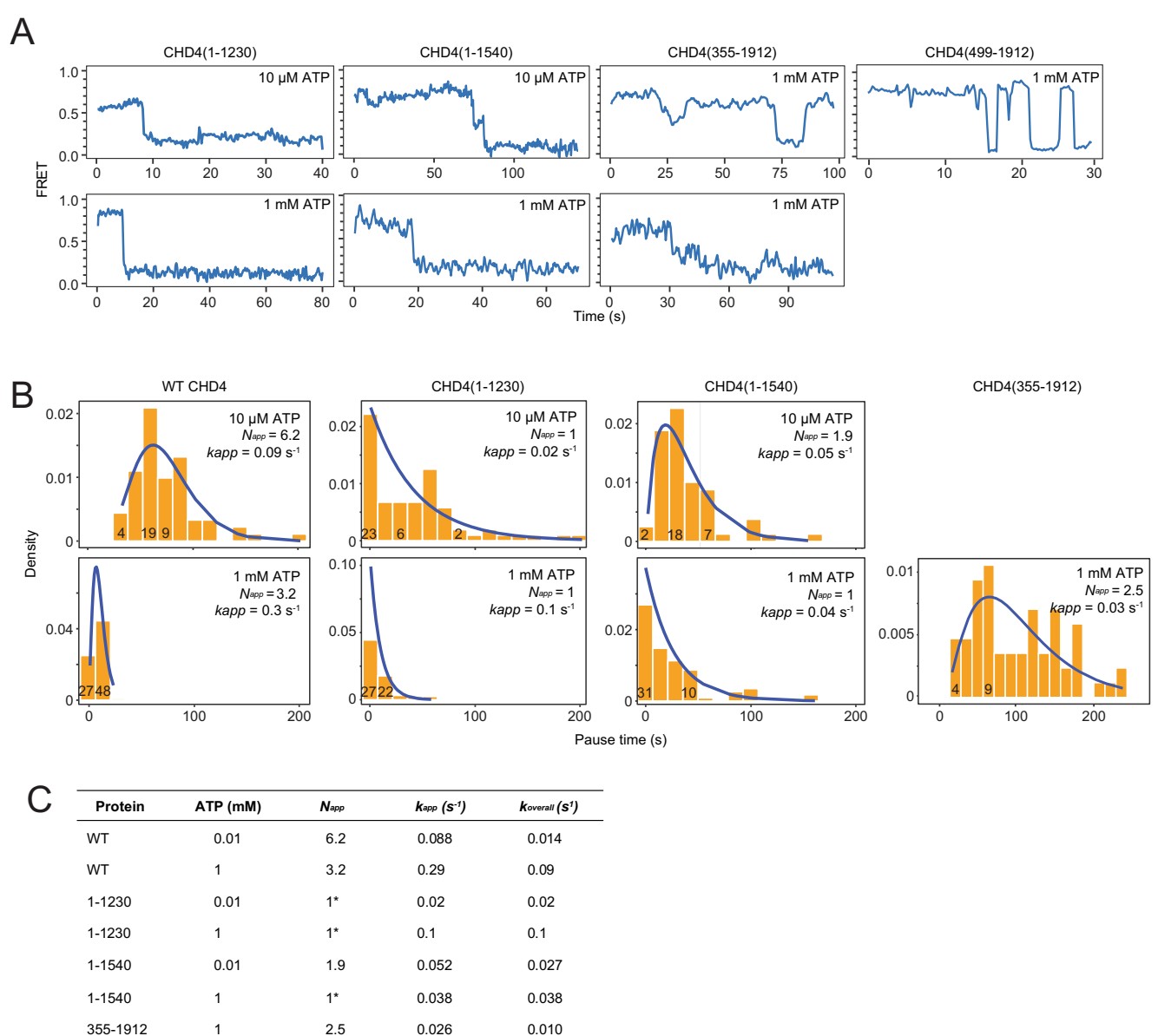

**Fig. 7 | smFRET data for CHD4 truncations. A** Typical smFRET traces for CHD4 truncations at the indicated ATP concentrations. **B** Histograms of the pause time at mid-FRET and fits in each case to a gamma or exponential function (as appropriate) in which the number of equivalent steps and the pseudo-first-order rate constant are fitted parameters. **C** Table summarizing parameters obtained from fits of the data in **B**. $k_{app}$ is the pseudo first-order rate constant for a single RDS. $k_{overall}$ is obtained from the ratio $k_{app}/N_{app}$ and represents the effective rate constant for the system to fully traverse $\tau_{pause}$ and eject ~5 bp of DNA from the exit side of the nucleosome. Source data for **A** and **B** are provided as a Source Data file.

expressed and highly abundant, might 'prime' chromatin for remodelling by partially separating DNA from histones[39]. In this role, the acidic and basic IDRs would bind histones and DNA, respectively, at the HMG binding site to ameliorate the effect of charge separation.

Our RTFA data show that deletion of residues 1–355 significantly reduces the nucleosome sliding activity of CHD4. This effect is likely due at least in part to the three-fold reduction we measured in substrate binding affinity. Strikingly, effects of similar magnitude on affinity and activity are observed upon simply halving the number of basic residues in IDR2, indicating that this region plays an important role in remodelling.

IDR2 shares sequence motifs with several other nucleosome-binding proteins. Figure 8A shows an alignment of the nucleosome binding domain (NBD) of HMGN2 with part of CHD4-IDR2. HMGN-family proteins decompact chromatin by acting in opposition to the

linker histone H1[40]. These ~100-residue intrinsically disordered proteins (which do not contain an HMG domain) share a 30-residue NBD, which is the sole determinant of the specific HMGN-nucleosome interaction. NMR analysis of the HMGN2-nucleosome interaction shows that a double arginine motif (Fig. 8A, *red*) directly contacts the H2A-H2B acidic patch (a contact point for many chromatin-associated proteins[41]), whereas a nearby PKPKK sequence (Fig. 8A, *blue*) contacts the nucleosome near the dyad[42]. CHD4-IDR2 harbours a sequence that features both motifs, suggesting that this region might make similar interactions with the nucleosome surface. The C-terminal tail of histone H1 also contains multiple repeats of the PKK motif (Fig. 8B). The H1 tail has been shown by cryoEM to contact linker DNA[43] and is essential for both high-affinity nucleosome binding and for the ability of H1 to condense chromatin into higher-order structures[44,45]. Condensation is opposed by HMGN proteins

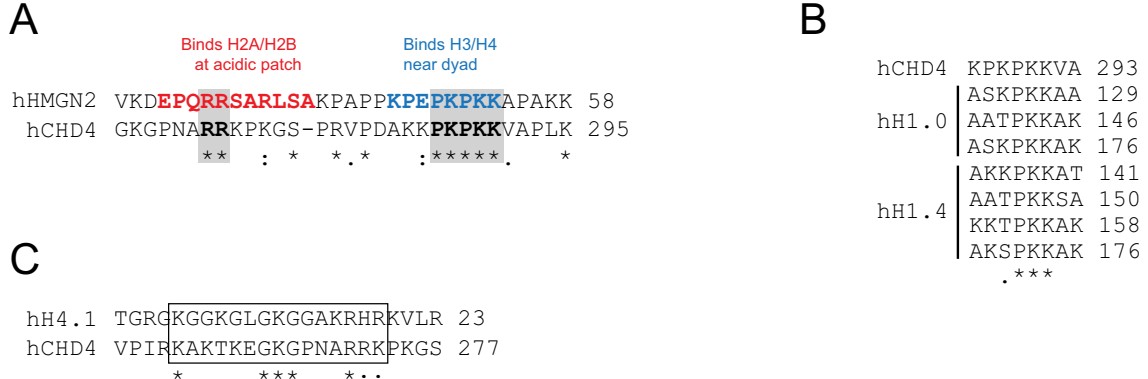

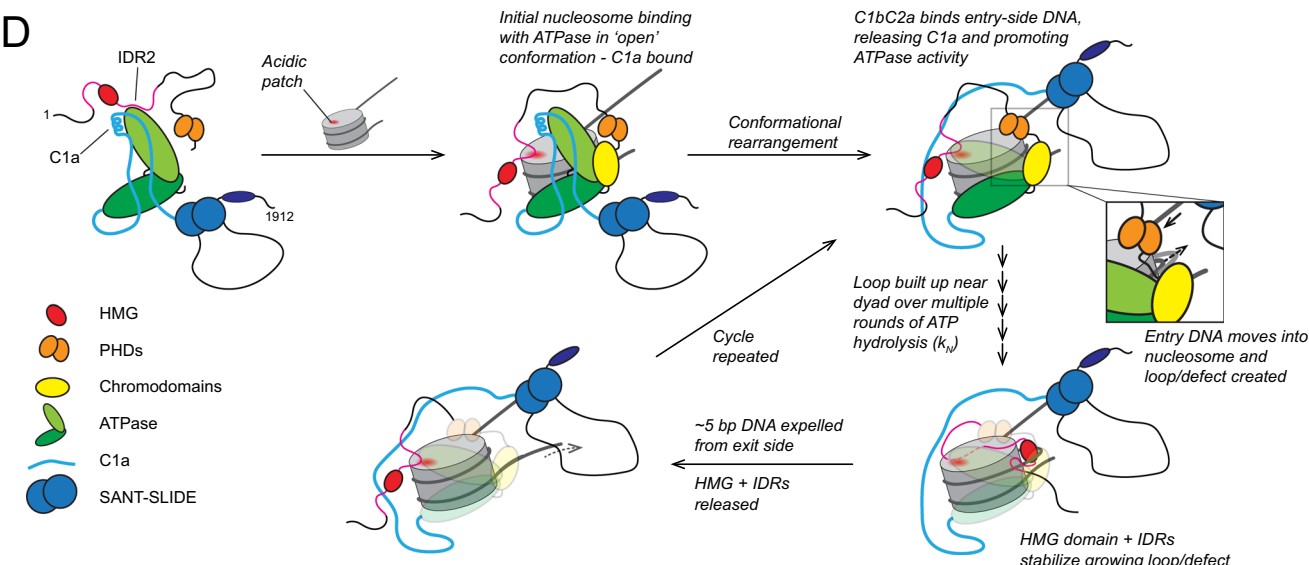

**Fig. 8 | Updated model for nucleosome remodelling by CHD4. A** Alignment of part of the nucleosome binding region of human HMGN2 (Uniprot: P05204) with the IDR of CHD4. **B** Alignment of a portion of CHD4 IDR2 with C-terminal repeats from the human linker histone H1 orthologues H1.0 (Uniprot: P07305) and H1.4 (Uniprot: P10412). **C** Alignment of the N-terminal tail of human histone H4.1 (Uniprot: P62805) with a portion of CHD4 IDR2. **D** Model for the mechanism of nucleosome remodelling by CHD4. Domains are coloured according to Fig. 1B.

Binding of CHD4 to a nucleosome involves contacts made by the PHD, chromodomains and translocase, as well as by IDR2, which contacts the acidic patch. In addition, the SANT-SLIDE domain contacts extranucleosomal DNA on the entry side, displacing the auto-inhibitory C1a domain and permitting cycles of ATP-driven DNA translocation to occur. As part of these cycles, the HMG domain (and perhaps the IDR1 and −2 domains) might assist in stabilizing the nucleosomal intermediates harbouring additional base pairs of DNA.

and, interestingly, HMGN proteins also suppress chromatin remodelling by the chromatin remodellers ACF and BRG1[46]. These observations point to an overlap in nucleosome binding mode between linker histones, HMGN proteins and chromatin remodelling enzymes, with short Arg- or Pro/Lys-rich motifs engaging conserved elements of nucleosome structure.

In line with this idea, the remodelling activity of both CHD4 and a range of other chromatin remodellers is reduced by mutation of the acidic patch[47,48]. In several cases, this effect is recapitulated by mutation of a specific cluster of arginine residues. For example, Arg-rich motifs in SNF2h and ALC1 engage the acidic patch and are required for SNF2h activity[49,50]. Similarly, three clustered arginines in in the N-terminal part of CHD7 promote remodelling activity and low-resolution cryoEM data show this region binding to the area that harbours the acidic patch[51]. CHD7 is also unable to effectively remodel nucleosomes in which the acidic patch is mutated. In other examples, the acidic patch promotes remodelling by binding auto-inhibitory Arg-rich sequences[29,48], underscoring the importance of the acidic patch in remodeller activity.

In chromatin, the acidic patch is also contacted by the basic N-terminal tail of histone H4 (residues -16–23) from adjacent nucleosomes[52,53], an interaction that promotes nucleosome stacking[54]. This same region of histone H4 binds to a conserved acidic pocket in the C-terminal lobe of the CHD4 translocase domain[10] and corresponding interactions are seen in structures of CHD1[5], SNF2[55] and ISWI[56]. Mutation or removal of this N-terminal tail increases the pause time markedly for ISWI-family remodellers[57–59], indicating that the tail functions not in substrate binding but in activation of ATP-dependent DNA translocation. The portion of IDR2 that harbours the RR motif also shows some sequence similarity to the H4 N-terminal tail (Fig. 8C).

Finally, it is notable that the increase in remodelling 'reversals' that we observe following deletion of residues 1–355 are mirrored in CHD7 when the three clustered arginine residues are mutated to alanine[51]. These observations point to a role for the N-terminal domain, and these arginines in particular, in regulating the integrity of DNA translocation.

Taken together, these data are consistent with a role for IDR2 in binding the acidic patch and/or extranucleosomal DNA and for the

N-terminal region more broadly in promoting and maintaining the integrity of nucleosome sliding activity in CHD4.

## A DNA-binding SANT-SLIDE domain relieves auto-inhibition by the C1a region

Our combination of fluorescence and gel-based remodelling assays and MST measurements demonstrates that the C1a region (residues ~1230–1380) significantly inhibits CHD4 activity. The extent of remodelling at equilibrium is ~six-fold less for CHD4(1–1380) than for WT CHD4. XLMS data point to a mode of inhibition mirroring that observed for CHD1, where the corresponding C-terminal bridge contacts both lobes of the translocase domain in an irregular conformation. Our XLMS data are consistent with this interaction involving the structurally equivalent surface of CHD4. A recent study of the *Drosophila* homologue of CHD4 also showed that a point mutation at the N-terminal end of the C1a region results in significantly higher remodelling activity[32], again indicating an inhibitory role for this region. Other remodellers also use auxiliary domains to hold the translocase in an inactive conformation; examples of such domains include the helical autoN domain of ISWI-family remodellers[29] and the macro domain of ALC1[60].

The abrogation of remodelling activity observed in CHD4(1–1380) is reversed when C1bC2a – or even just C1b – is restored in CHD4(1–1810) and CHD4(1–1540), respectively (Fig. 4A). Furthermore, our X-ray crystal structure of C1bC2a reveals a SANT-SLIDE domain that closely resembles the structure observed in both CHD1 and ISWI. MST measurements show that C1bC2a is able to bind nucleosomal DNA with moderate affinity and all CHD4 C-terminal truncations, including the hyperactive CHD4(1–1230), display lower affinities for nucleosomes than WT, consistent with a role for C1bC2a in substrate recognition.

It has been shown that binding of the SANT-SLIDE domain of CHD1 to entry side DNA relieves the inhibition imposed by the C-terminal bridge and that, in the absence of entry-side DNA, CHD1 is unable to slide nucleosomes[61]. Our data on CHD4 are consistent with the same mechanism of action. It is notable that the SANT-SLIDE domain of CHD4 was not observed in the recent cryoEM structure of CHD4 bound to a nucleosome[10]; however, the nucleosome used in this work carried only 4 bp of DNA at the entry side, suggesting that it might have been too dynamic in the absence of its cognate binding site to be observed. Consistent with the idea that this domain does not take up a defined conformation relative to the translocase domain in the absence of substrate, our XLMS data show crosslinks between the C1bC2a domain and the translocase domain that are best satisfied by multiple relative orientations of the two regions. We also note that a recent structure of the human ISWI enzyme SNF2h bound to a 0W60 nucleosome did not resolve the HAND-SANT-SLIDE domain, despite the presence of extranucleosomal DNA[62], suggesting that the interaction with DNA could be transient, thus allowing the (HAND)-SANT-SLIDE domain to adjust its binding site as remodelling progresses.

Our data redefine the domain boundaries of the C-terminal "domain of unknown function" of CHD4. The observation that only CHD4(1380–1810), out of many CHD4 C-terminal constructs tested in our laboratory, can be expressed and purified as a folded and functional polypeptide supports this domain definition. The DNA binding capacity of this domain suggests that it could relieve the C1a-imposed auto-inhibition by interacting with linker DNA, and thus activate remodelling. This proposed mechanism could explain the observation of mutations in the 1230–1810 region that are frequently observed in patients with breast cancer[63] or neurodevelopmental disorders[19,63].

## Mechanistic convergence of ISWI and CHD chromatin remodellers

Since the discovery of multiple classes of chromatin remodelling enzymes (*e.g.*, INO80, ISWI, SWI/SNF and CHD), a substantial body of biochemical analysis has sought the mechanisms by which these enzymes act and the identification of similarities and differences between the classes. Recent structural analysis has demonstrated significant commonality in the DNA translocase domains of these enzymes and in how they engage a single nucleosome[5,6,56,64]. This structural similarity strongly hints at mechanistic similarity but the very different domain structures of the different remodeller classes, together with the fact that each operates in the context of very different molecular assemblies, has made it more challenging to compare and contrast their mechanisms of action.

Clear similarities have been described between CHD1 and members of the ISWI family. ISWI proteins are conserved across Eukarya and in humans there are two ISWI proteins, SNF2H and S2L, which are found in at least seven distinct remodelling complexes (reviewed in[65]). ISWI proteins and CHD1 share DNA-binding SANT-SLIDE domains in addition to their DNA translocase domains and, for both proteins, two negative regulatory elements have been defined that appear to play similar roles. What has not been clear is how closely the mechanism of action of CHD1 resembles that of the other eight CHD family members. A SLIDE domain has been predicted in CHD7–9 but never demonstrated experimentally, whereas the presence of a SANT domain in CHD3–5 has been occasionally though not consistently suggested. Our data demonstrate that close mechanistic similarities exist between CHD4 and CHD1. Furthermore, sequence comparisons and AlphaFold predictions make it clear that CHD2–9 are all very likely to harbour the same SANT-SLIDE module (Supplementary Fig. 6), underscoring this probable mechanistic commonality.

Finally, we suggest a more complete model for nucleosome sliding by CHD4—and for other CHD proteins (Fig. 8D). In the free state, the ATPase activity of CHD4 is inhibited by a combination of IDRs and the C1a region binding to the translocase domain. Engagement with the nucleosome releases both of these inhibitory regions: the former binds to the acidic patch and the latter is allosterically released from the translocase through binding of the SANT-SLIDE module to entry-side linker DNA. Successive binding and hydrolysis of 5–6 ATP molecules causes the build-up of a defect or loop of partially translocated DNA in the nucleosome (perhaps near the dyad, stabilized by the IDR1-HMG-IDR2 unit), until these base pairs are released in a concerted fashion from the exit side. This process can be repeated multiple times, allowing CHD4 to effectively slide the histone octamer along DNA in a processive manner. Our smFRET analysis of the CHD4(1–1230) mutant confirms that loss of the C-terminal region increases the rate of the multiple ATP-dependent DNA translocation steps, consistent with the auto-inhibition model.

## Methods

### Histone expression and purification

Human histones were expressed and purified using HPLC as described in[4]. Briefly, pET28a plasmids containing tagless human H2A, H2B, H3.1 and H4 genes were expressed in *Escherichia coli* (*E. coli*) BL21 (DE3). All histones, except for H4, were expressed overnight at 37 °C using ZYM-5052 auto-induction media[66]. H4 culture was expressed in 2× yeast extract tryptone (YT) media and induced with 1 mM isopropyl ß-D-1-thiogalactopyranoside (IPTG) at $OD_{600}$~0.8–1.0 and expressed overnight at 37 °C. After sonicating the cell pellets in 50 mM Tris-HCl pH 7.5, 100 mM NaCl, 1 mM beta-mercaptoethanol (BME) and clarification at 15000 × g for 30 min, and the insoluble fraction was washed twice with lysis buffer containing 1% (v/v) Triton X-100, and then twice without Triton X-100. The pellet was dissolved in 10 mL of unfolding buffer (20 mM Tris-HCl pH 7.5, 6 M guanidinium HCl and 1 mM dithiothreitol (DTT)) per L of culture by stirring at room temperature overnight. After clarification, filtered supernatants were injected onto a preparative Vydac protein and peptide C18 column (300-Å pore size, Catalogue No. 218TP1022) at a flow rate of 7 mL/min (20–70% acetonitrile over 40 min, 0.1% TFA). The fractions containing the target

protein (as judged by liquid chromatography-mass spectrometry (LC-MS) analysis) were freeze-dried and stored at −20 °C in sealed containers.

Labelled H2A for RTFA and smFRET experiments was prepared in the same way as in[4], except AF488-maleimide was used for nucleosomes in RTFA. The purified histones were folded into octamer using the method in[67].

### Nucleosomal DNA preparation
Nucleosomal DNA production are as described in[4]. Briefly, Widom 601.2 sequences was PCR amplified from plasmid with different primers in order to generate different lengths of flanking DNA. Fluorescence tagged primers were used to produced labelled DNA. The PCR products from 5- or 10-mL reaction were pooled, and ethanol precipitated. The DNA was then extracted from the redissolved pellet using phenol:chloroform:isoamyl alcohol (25:24:1) mixture, followed by a chloroform wash to remove residual phenol. The aqueous layer containing DNA was then concentrated by ethanol precipitation and separated on a 5% TBE polyacrylamide gel at 150 V for 40 min in the cold room. DNA in the correct band was cut from the gel and electroeluted, followed by concentrating with ethanol precipitation.

### Nucleosome reconstitution
Details for octamer refolding and nucleosome reconstitution are given in[4]. Folded histone octamer and nucleosomal DNA were mixed in 1:1 molar ratio and dialysis against NaCl gradient using a double dialysis method[68]. The octamer and DNA mxiture in 10 mM Tris-HCl pH 7.5, 2 M NaCl, 1 mM EDTA was loaded into a small dialysis button, which was then placed into a dialysis bag containing 30 mL of the same buffer. The bag was then dialysed against 2 L of 1× TE buffer overnight at room temperature.

The next day, the dialysis button was dialyzed further against 10 mM Tris pH 7.5, 2.5 mM NaCl and 0.1 mM DTT. Content in the dialysis button was harvested and the nucleosome quality was checked by gel electrophoresis on a 5% TBE polyacrylamide gel.

### Expression and purification of CHD4 in mammalian cells
All CHD4 constructs used for activity assays were expressed in HEK293 Expi cells. The culturing and expression conditions are stated in[4]. CHD4 variants used in this study were cloned into pcDNA3.1 vector with a FLAG tag. The sequences of used for CHD4(227-362) modifications are given in Supplementary Fig. 2.

For each construct, 30–50 mL of culture was transfected and after 65 h incubation at 37 °C with 5% CO$_2$ and horizontal orbital shaking at 130 rpm, the cells were harvested at 500× g for 10 min and washed twice with 1× PBS.

To purify the FLAG-tagged protein, pellet from 1 mL culture was sonicated in 0.5 mL of lysis buffer (50 mM Tris/HCl, 500 mM NaCl, 1% (v/v) Triton X-100, 1× cOmplete EDTA-free protease inhibitor (Roche, Basel, Switzerland)), 100 mM ATP, 0.2 mM DTT, pH 7.9, and then clarified the lysate via centrifugation (≥16,000 × g, 20 min, 4 °C). Per 15 mL culture, 100 µL anti-FLAG Sepharose 4B beads (Biomaker, Houston, USA, catalog #23102) was added to the cleared lysate. The mixtures were incubated overnight at 4 °C with orbital rotation. Post-incubation, the beads were washed with 3× wash buffer A (50 mM HEPES pH 7.5, 500 mM NaCl, 0.5% (v/v) IGEPAL, 3 mM ATP, 3 mM MgCl$_2$, 0.2 mM DTT) and then twice with wash buffer B (50 mM HEPES pH 7.5, 150 mM NaCl, 0.2 mM DTT). Bound proteins were eluted by 5 × 300 µL treatment with 'elution' buffer (10 mM HEPES, 150 mM NaCl, 150 µg/mL 3× FLAG peptide (MDYKDHDGDYKDHDIDYKDDDDK), pH 7.5), each with 0.5 h incubation at 4 °C. Concentrations of eluted CHD4 were estimated using densitometry (ImageJ) by loading onto SDS-PAGE along with a known amount of BSA and staining with SYPRO® Ruby. The quality of purified CHD4 constructs is shown in Supplementary Fig. 7.

### Expression and purification of C1bC2a constructs
CHD4-C1bC2a$_{short}$ [1380-1551 (C1b) joined to 1690–1810 (C2a)] was expressed as a GST-fusion protein in pGEX-6P. Protein overexpression was induced in *E. coli* BL21 (DE3) cells by 0.4 mM IPTG and cells grown for 20 h at 20 °C in Luria broth. Cell pellets were sonicated in lysis buffer containing 50 mM Tris-HCl pH 7.6, 500 mM NaCl, 0.1% Triton, 5 mM BME, 1 mg/ml lysozyme, 1 mM phenylmethylsulfonyl fluoride, 1× cOmplete® protease inhibitor (Roche). After clarification (30 min at 20,000 × g at 4 °C), lysate was incubated with GSH-Sepharose resin for 2 h at 4 °C, then washed with high (50 mM Tris pH 7.4, 500 mM NaCl, 1 mM DTT) and low (50 mM Tris pH 7.4, 150 mM NaCl, 1 mM DTT) salt buffers. Beads were incubated with HRV-3C overnight at 4 °C. Cleaved CHD4-C1bC2a was further purified by size exclusion chromatography (SEC) on a Superdex 75 10/300GL column (GE Healthcare) and eluted with 20 mM HEPES pH 7.6, 150 mM NaCl and 1 mM TCEP. After protease cleavage, an additional non-native sequence (GPLGS) remained at the N-terminus of CHD4-C1bC2a.

### Protein crosslinking, mass spectrometry and data analysis
For the crosslinking experiment, 25 µg of protein was used. Bis-sulfosuccinimidyl suberate (BS3; 25 mM stock solution in Milli-Q water) was added to a final concentration of 1 mM and allowed to react for 30 min at 37 ºC. Post-crosslinking, excess BS3 was quenched with a final concentration of 20 mM Tris at 37 ºC for 20 min then snap-frozen in liquid nitrogen and freeze-dried.

Crosslinked sample trypsinisation and peptide size exclusion chromatography were performed essentially as described previously[69]. Briefly, dried, crosslinked samples were resuspended in 8 M urea to give a final concentration of 0.5 mg/mL protein. Samples were then reduced (5 mM TCEP, 37 °C, 30 min) and alkylated (5 mM iodoacetamide, 20 min, room temperature in the dark). The samples were then diluted to 6 M urea with 50 mM Tris-HCl pH 8 and Trypsin/Lys-C mix (Promega) was added to an enzyme:substrate ratio of 1:25 (w/w) and incubated at 37 °C, 4 h. Following this, the samples were further diluted to 0.8 M urea with 50 mM Tris-HCl pH 8 and the sample was further incubated at 37 °C overnight (16 h minimum). After the overnight digestion, the samples were acidified with formic acid to 2% (v/v) and centrifuged at 16,000 g for 10 min. The supernatant was then desalted using a 50-mg Sep-Pak tC18 cartridge (Waters), and eluted in 60:40:0.1 acetonitrile:water:formic acid snap-frozen in liquid nitrogen and dried in a freeze-dryer.

For SEC fractionation, the dried desalted peptides were resuspended in 250 µL of SEC mobile phase (acetonitrile:water:trifluoroacetic acid, 30:70:0.1 (v/v/v)) and separated on a Superdex Peptide HR 10/30 column (GE Healthcare). A maximum of 2 mg of peptide was injected onto the column per SEC run. A flow rate of 0.5 mL/min was used and the separation was monitored by UV absorption at 215, 254 and 280 nm. Fractions were collected as 0.5-mL fractions. Based on the UV absorption traces, fractions of interest (retention volumes ~9–13 mL) were pooled, snap-frozen and freeze-dried.

Following SEC, peptides were subjected to high-pH reverse phase fractionation. Dried peptides from the SEC step were resuspended in Buffer A (3% (v/v) acetonitrile, 5 mM ammonium formate, pH 8.5) and loaded onto a 50-mg Sep-Pak tC18 cartridge (Waters), washed with Buffer A and eluted with increasing amounts of Buffer B (80% (v/v) acetonitrile, 5 mM ammonium formate, pH 8.5) in a stepwise manner (10% steps from 10–90% B). Fractions were collected for each elution step and concatenated into three pools; Pool 1 comprised of 10%–40%–70% B elutions, Pool 2 comprised of 20%–50%–80% B elutions, and Pool 3 comprised of 30%–60%–90% B elutions. Pooled fractions were then freeze-dried.

Dried peptides were resuspended in 4% (v/v) acetonitrile, 0.1% (v/v) formic acid and loaded onto a 30 cm × 75 µm inner diameter column packed in-house with 1.9 µm C18AQ particles (Dr Maisch

GmbH HPLC) using a Dionex UltiMate 3000 UHPLC (ThermoFisher Scientific). Peptides were separated using a linear gradient of 10–50% buffer B over 108 min at 300 nL/min at 55 °C (buffer A consisted of 0.1% (v/v) formic acid; while buffer B was 80% (v/v) acetonitrile and 0.1% (v/v) formic acid) and were analysed on a Q-Exactive HF-X mass spectrometer. The following MS protocol was used: Following each full-scan MS1 at $R = 60,000$ at 200 $m/z$ (350 – 1800 $m/z$, AGC = $3 \times 10^6$, 100 ms max injection time), up to 12 most abundant precursor ions were selected MS2 in a data-dependent manner (HCD, $R = 15,000$, AGC = $2 \times 10^5$, stepped NCE = (25, 30, 35), 25 ms max injection time, 1.4 $m/z$ isolation window, minimum charge state of +4; dynamic exclusion of 20 s). The XLMS experiment was performed once.

Crosslinked peptides were identified using pLINK2 v2.3.9[70]. Key pLINK search parameters were as follows: peptide mass between 600–10,000 Da and peptide length between 6–100 were considered, precursor mass tolerance ±15 ppm, product-ion mass tolerance ±20 ppm, enzyme specificity of trypsin with up to two missed cleavages (excluding the site of crosslinking) per chain. Allowed variable modification = oxidation (M), fixed modification = carbamidomethyl (C), BS3 crosslinker settings: crosslinking sites were Lys, Ser, Tyr, Thr and protein N-terminus, crosslink mass-shift 138.068 Da.

The search database used was the Uniprot human reference proteome (UP000005640; May 2020; 20,286 entries). An FDR of 1% was used. Only peptides with a precursor mass error of ≤±10 ppm and with at least four fragment ions on both the alpha- and beta-chain each were retained for further analysis. XLs that were identified as possibly either inter-protein or intra-protein (due to isoform sequence similarities) were considered as intra-protein as the likelihood of intra-protein XLs is higher than inter-protein XLs.

The final non-redundant list and unfiltered pLINK2 search outputs can be found in Supplementary Data 1. All mass spectrometry data and XLMS search results have been deposited to the ProteomeXchange Consortium via the PRIDE partner repository[71] with the dataset identifier PXD033633.

## Crystallization and structure determination

Initial crystallization screening was performed with commercial sparse matrix screens using Mosquito Robot (TTP Labtech Ltd.) Crystallization trials were set up in 96-well MRC 2 plates (Hampton Research) at 4 °C in sitting drop format using 0.2 μL of CHD4-C1bC2a protein solution (12 mg/mL, in 20 mM HEPES pH 7.6, 150 mM NaCl, 1 mM TCEP) and 0.1 μL of crystallization solution. The initial crystallization hit was obtained with the Morpheus screen (Molecular Dimensions) in 0.1 M bicine/Trizma base pH 8.5, 10% w/v PEG 8000, 20% v/v ethylene glycol, and 0.1 M mixture of carboxylic acids. Further optimization was carried out using the Additive screen (Hampton Research) and in-house grid screen in both sitting and hanging drop vapour diffusion format. This resulted in single crystals in 0.1 M bicine/Trizma base pH 8.3, 10% w/v PEG 8000, 20% v/v ethylene glycol, 0.1 M mixture of carboxylic acids, and 20 mM spermidine at 4 °C. The crystals were harvested into a cryoprotectant solution containing 25% glycerol in mother liquor before cryocooling in liquid nitrogen.

X-ray diffraction data sets were recorded on MX2 beamline (microfocus) at the Australian Synchrotron at 100 K and a wavelength of 0.9537 Å. CHD4c1bc2a crystals diffracted to a resolution of 2.9 Å. Indexing and integration of the data was performed using XDS[72]. An AlphaFold model of CHD4-C1bC2a from ColabFold[73] was used as a search model in the MRSAD protocol of Auto-Rickshaw[74] to perform automated phasing. The initial structure was partially refined to an $R_{free}$ of ~32% with REFMAC[75] using Auto-Rickshaw pipeline. The crystal belonged to space group P 1 21 1, with two molecules in the asymmetric unit. Further model building was performed by iterative rounds of manual model building in real space, COOT[76], followed by refinement using PHENIX[77]. Final refinement rounds were performed in REFMAC and PHENIX, keeping X-ray/stereochemistry weight, refining TLS parameters and individual B-factors, and manually inspecting the model at each step to examine quality-of-fit to the electron density. Because of the occurrence of disordered regions and intrinsic flexibility, including long loops within the structure, the structure was refined at best to an $R_{work}/R_{free}$ of 26.3/30.1%. Careful attention was paid to the local indicators of structure quality, which are important for the accurate biological interpretation of structures[78,79]. The quality of the final model was validated using Molprobity and the structure was submitted to the PDB (PDB ID: 8D4Y). Crystallographic details and statistics are listed in Supplementary Table 1. Our statistics at 0% and 0.3% for Ramachandran and sidechain outliers respectively indicate a reasonable refinement at 2.9 Å, given the experimental data.

## Gel-based remodelling assay

Each remodelling reaction contained 60 nM Cy3-labelled 0w60 nucleosomes, 1 mM ATP and 10 nM remodeller in 50 mM Tris pH 7.5, 50 mM NaCl, 3 mM MgCl$_2$. The reactions were incubated at 37 °C and then stopped by placing them on ice and the addition of 0.5 μg salmon sperm DNA or competitor DNA and 4% (w/v) sucrose prior to electrophoresis on 5% polyacrylamide gels. Gels were imaged on an FLA-9000 laser scanner.

## Real-time fluorescence assay (RTFA)

Nucleosomes assembled from $0^{BHQ1}$w60 DNA and AF488-labelled octamer were used for RTFAs. The reaction mixtures (typically 50 μL) containing 60 nM nucleosome, 10 nM remodeller in 50 mM Tris-HCl pH 7.5, 50 mM NaCl and 3 mM MgCl$_2$ were loaded into the wells of a Corning® black nonbinding surface half-area 96-well plate, which was then subsequently put into a Tecan® Infinite M1000 Pro plate reader pre-warmed to 37 °C. Reactions were monitored at 526 nm for 5 min prior to the addition of 1 mM ATP by the injector unit of the plate reader, and the fluorescence changes were monitored at 37 °C for another 25 min, with data points recorded every 30 s. Each trace shown is an average of three technical replicates normalised with a nucleosome-only control. Data were fitted to an asymptotic exponential equation:

$$y = F_{lim}(1 - \exp(-k_{RTFA}*t)) \tag{1}$$

where $F_{lim}$ is the limiting value of the fluorescence, $k_{RTFA}$ is a pseudo-first-order rate constant (units of min$^{-1}$), and $t$ is the time in minutes after addition of CHD4 and ATP.

## Single-molecule FRET assay

The instrument and protocol used for single-molecule FRET assays were as reported previously[4]. An Olympus IX-71 based model was modified to build an objective type total internal reflection fluorescence (TIRF) microscope to record single-molecule movies. A coherent Sapphire green (532 nm) laser was used to excite donor (AF555) molecules by focusing onto a ×100 oil immersed objective and scattered light was removed using a 560-nm long pass filter. Donor and acceptor (AF647) signals were collected at 565 and 665 nm using a band pass filter (560–600 nm and a long pass filter at 650 nm, respectively). Then, both signals were first split by a 638-nm dichroic mirror using Photometrics Dual View (DV-2) and then were focused onto a CCD camera (Hamamatsu C9 100-13), simultaneously. The camera was controlled by the provided software and micromanager 2.0 beta. Single-molecule movies were recorded at 5 frames per second. Nucleosomes labelled with both fluorophores as described above were immobilized in microfluidic channels on neutravidin coated coverslips and imaged in a TIRF microscope following excitation of donor (AF555) fluorophores. Movies of both donor and acceptor (AF647) signals were collected (565 and 665 nm, respectively) at a frame rate of 5 s$^{-1}$ at room temperature (20 ± 1 °C) for 3–5 min

(until bleaching of the acceptor dye). For remodelling assays, additional ATP (0.01–20 mM) was included in the imaging buffer.

Data analysis was carried out in MATLAB as described previously[4]. Because the acceptor dye was on histone H2A, three classes of nucleosomes were observed. Mid-FRET nucleosomes had the acceptor dye proximal to the donor dye on the DNA, low-FRET nucleosomes had the acceptor dye distal to donor dye, and high-FRET nucleosomes displayed an acceptor dye on both H2A subunits. Typically, data could be modelled by three Gaussians with mean FRET values of 0.78, 0.67 and 0.42 for exit side 0w60 nucleosomes. For our analysis, we chose molecules showing a mean FRET value in the range of 0.55 to 0.7 and displaying a single acceptor photobleaching step.

To measure the rate of remodelling reaction at different ATP or CHD4 concentrations, values of $\tau_p$ were measured from multiple traces. Histograms of $\tau_p$ were fitted to a gamma distribution:

$$f(t) = \frac{k^N t^{N-1}}{\Gamma(N)} \exp(-kt) \tag{2}$$

where $N$ is the number of identical steps that make up the rate-determining step for remodelling and $k$ is the rate of a single step.

### Microscale thermophoresis (MST)

All CHD4 constructs were concentrated to ~2–3 μM and buffer exchanged into dilution buffer (50 mM HEPES pH 7.5, 50 mM NaCl, 3 mM MgCl$_2$) using Amicon Ultra-0.5 mL Centrifugal Filters (30 or 100 K MWCO, Merck Millipore). After re-measuring the concentrations by densitometry, a set of titration points for each protein was prepared by 1:2 serial dilution using the same dilution buffer. The diluted CHD4 were then mixed with 50 mM HEPES pH 7.5, 100 ng/μL BSA, 0.5% (v/v) glycerol and 25 nM Cy5-labelled 46w60 nucleosome (or naked DNA). We chose 46w60 instead of 0w60 because there are two CHD4 binding sites on each nucleosome (at the diametrically opposed superhelical location 2 sites SHL2 and SHL−2) and adjacent linker DNA enhances binding to the corresponding site[4]. The additional 46 bp serves to equalize the affinity for the two sites, simplifying analysis.

WT-CHD4-C1bC2a was concentrated to 70 μM in 20 mM HEPES pH 7.4, 150 mM NaCl, 1 mM TCEP and 1:2 serial dilution samples were prepared in the same buffer. CHD4-C1bC2a samples were mixed with Cy5-labelled 46w60 nucleosomal DNA to a final concentration of 25 nM, 20 mM HEPES pH 7.4, 120 mM NaCl, 100 ng/μL BSA, and 1 mM TCEP. Following 10 min incubation on ice, the samples were loaded into premium coated capillary tubes (Nanotemper) before undergoing MST in a Monolith™ NT.115 instrument. Thermophoresis was conducted at 80% LED power, and 20% MST power. Thermophoresis data were then analysed with MO.Affinity Analysis software v2.3.

### Reporting summary
Further information on research design is available in the Nature Portfolio Reporting Summary linked to this article.

## Data availability
The data that support this study are available from the corresponding authors upon reasonable request. The XLMS data generated in this study have been deposited in the ProteomeXchange database under identifier code PXD033633. The crystal structure of C1bC2a-Δ139 was submitted to the PDB (PDB ID: 8D4Y). Source data are provided with this paper. Data (without normalization) underlying Figs. 2C–D, 3A–D, 4A, B, 5C, 6B, C, 7A, B, Supplementary Fig. 5, as well as original gel pictures in Fig. 2A and Supplementary Fig. 7 are provided in the Source_Data.xlsx file. All smFRET traces used for analysis in Figs. 6 and 7 are available in smFRET_source_data folder. Both.xlsx file and the folder are in the Source data folder. Source data are provided with this paper.

## Code availability
Custom code scripts used for data analysis in IDL and MATLAB program are publicly available at https://cplc.illinois.edu/software/. Script used for reaction step simulation in Supplementary Fig. 5 is available on Zenodo via public permanent https://doi.org/10.5281/zenodo.7306124.

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

## Acknowledgements

This work was funded by a National Health and Medical Research Council grant to JPM and AMvO (APP1126357). This research was undertaken in part using the MX2 beamline at the Australian Synchrotron, part of ANSTO, and made use of the Australian Cancer Research Foundation (ACRF) detector. We acknowledge the Sydney Mass Spectrometry Core Research Facility at the University of Sydney for providing access to mass spectrometers and thank the technical staff for the maintenance of the instruments.

## Author contributions

Conceptualization, Y.Z., B.P.P., A.M.O. and J.P.M.; Methodology, Y.Z., B.P.P., H.M.S., J. K.K.L., A.P.G.S., X.J.R. and M.J.B.; Software, B.P.P., C.D., and S.P.; Investigation Y.Z., B.P.P., H.M.S.; Resources, A.M.O. and J.P.M.; Writing – Original Draft, Y.Z. and J.P.M.; Writing – Reviewing and Editing, Y.Z., S.M., A.M.O. and J.P.M.; Supervision, A.M.O. and J.P.M.

## Competing interests

The authors declare no competing interests.
