## [Peer Review File · Nature Communications]

REVIEWER COMMENTS

Reviewer #1 (Remarks to the Author):

In this manuscript, the authors studied the role of auxiliary domains in modulating CHD4 activity and found that the N- and C-terminal auxiliary domains play different roles in CHD4 remodeling activity. Further studies indicated that an N-terminal intrinsically disordered region (IDR) promotes remodeling integrity in a manner depending on the composition but not sequence of the IDR, while the C-terminal region harbours an auto-inhibitory region which is relieved by a previously unrecognized C-terminal SANT-SLIDE domain.

Overall, the validity of approach and quality of the experimental work is of a high standard. The assays presented are all relevant to the core question of this manuscript and the presentation is focused and concise.

I only have a couple of issues that can be considered for further improvement.

Major:

1. In the "The N-terminal intrinsically disordered region regulates remodelling activity" section, what is the reason of "a N-terminal truncation that completely lacks IDR1 – CHD4(145–1912) – is even more active than WT (Figure 3B)."? Does this mean that the IDR1 is also an auto-inhibitory motif of the activity of CHD4? Furthermore, the activity comparison between IDR1 mutants and CHD4(355-1912) in the same section is not appropriate, since both the HMG and IDR2 except IDR1 were deleted in CHD4(355-1912).
2. There are several comparisons of different truncations on nucleosome binding activity and remodeling behaviour in second paragraph on page 5, so what is the relationship between the substrate binding affinity and activity of CHD4? For most enzyme, the stronger substrate binding affinity, the higher activity. Is it fit for CHD4? If not, why?
3. The binding affinity to 46w60 DNA of C1bC2a [CHD4(1380–1800)] is significant weaker than the WT CHD4 to 46w60 DNA (10 μ M vs 350 nM), how can "the SANT-SLIDE domain acts to sense the presence of a nucleosomal substrate"?

Minor:

1. Figures numbering should be done according to the order of results, for example, figure 3D should swap with 3C, so 3C following 3B not 3D; figure 4A should also be cited in "Deletion of the C-terminal 102 residues..... three-fold (Figure 4B, Supplementary Figure 1)."; the MST data should be at the lowest position in supplementary figure 1, and so on.
2. Typo, it should be "remodeling" but not "remodelling"; "an N-terminal truncation" but not "a N-terminal truncation"; "the region C1a to C-terminal (residues 1340–1912)" but not "the region C-terminal to C1a (residues 1340–1912)".
3. Carefully comparison indicated that the MST curve of WT CHD4 in figure 4B is same as that in figure 3E, so why there is difference of the scale or digital value of the y-axis?
4. The label of the same information in different figures should be same, such as WT CHD4 in figures 3E and 4B, Napp and steps in figures 6C and 7B...
5. What does the blank region mean in supplementary figure 1 of the MST binding data of IDR1 mutants? Furthermore, the unit of the KD should be nM but not μ M.
6. The structural data need to be further refined, since the R values are so high and the gap between Rwork/Rfree is so big. Furthermore, there are many mistakes in structural table, such as the description of the unit cell, the format of CC1/2....

Reviewer #2 (Remarks to the Author):

In this manuscript, Zhong and co-workers explore the role of auxiliary domains in modulating CHD4 activity. CHD4 is an essential chromatin remodeler that repositions histone octamers on chromatin. But some of its structure doesn't have well-characterized functions. The authors employ an array of

approaches to investigate these functions, including fluorescence, gel-based assays, single molecule spectroscopy and cross-linking mass spectrometry. They find that the N-terminal disordered region promotes remodelling integrity and that the C-terminal region has auto-inhibitory activity regulated by the C-terminal SANT-SLIDE domain and DNA binding. Overall, the paper is clearly written and of interest to the general readership of Nat. Coms. The results are novel and exciting. There are some points that the authors should be able to address prior to acceptance:

Major Issues:

- Figure 2A: the gel is of poor quality and makes it challenging to assess the authors' conclusions. Do the dashed lines indicate where gels have been cut and combined? Was the same amount loaded in each lane? By eye, it looks like the total amount in each lane varies. For lanes 2 and 3 (WT CHD4 and 1-1230), the gel shows roughly an equal split between slid and non-slid at equilibrium. However, in Fig 2C, the real-time fluorescence assays show that more nucleosomes slid by 1-1230 compared to WT. Therefore, a reader might expect to see more in the upper band for 1-1230 in the gel. Could the authors please redo the gel, either loading more or imaging for longer such that it is easier to see the bands? Using a fluorescently labelled nucleosome such as is used in Fig 6 and Fig 7 should make it easy to visualise the bands.

- Figure 2C and D: In 2C, the max fluorescence reached is ~200 AU. In Figure 2D (which appears to be the fit of data from 2C), the Y axis shows a maximum of ~20 AU, while the bar itself is labelled with the number 2100. Could the authors please clarify these different values?

- Figure 2C: the figure caption states: "t is the time in minutes after addition of CHD4 and ATP". Could the authors clarify if CHD4 and ATP were added simultaneously? From Fig 2C, the dashed line indicates that only ATP was injected. The materials and methods section also has conflicting information on whether CHD4 and nucleosome were preincubated together before the addition of ATP or if ATP and CHD4 were added at the same time.

- Figure 2D: the figure caption says that the rate and max fluorescence values were determined by fitting to a logarithm function. Why was a logarithm function chosen here and an exponential function in Fig 2C?

- Figure 3C right panel: The fluorescence increase for both the WT and del- mutant appear to be biphasic in this experiment, with a fast phase and a slow phase. Elsewhere the WT fluorescence increase looks to be single exponential. Could the authors comment on this, please?

- In the text, when describing the Kd values determined by MST, it would also be useful to state the error in each value.

- Figure 4A: The data presented here for the 1-1230 truncation looks like it could be the same data as is presented in Fig 2C for the same truncation. Is this the exact same data, and are the authors including it again for comparison? If so, it should be stated clearly in the caption that this data is duplicated from a different figure for the purpose of ease of comparison. Could the authors please confirm that no other data has been presented in multiple panels without explicitly stating such?

- Figures 6 and 7: For the example FRET trajectories given, it is hard to judge the FRET value without any additional y-axis ticks. In the methods section, it is stated that molecules displaying a FRET value between 0.55 and 0.7 were included in the analysis (proximal labelled molecules). Judging by eye, many of the example trajectories shown (particularly in Fig 7A) start at a FRET value that looks >0.7. Are the authors confident that these high FRET molecules are proximal labelled and not double labelled molecules?

Minor Issues:

- Page 2, paragraph 3: it may read better if the word "formed" was replaced with "form" in the sentence beginning, "The translocase domain is immediately preceded by..."
- For the CHD4-IRD2-del+ mutant, the IRD2 region is replaced with (almost the entire length of) alpha-synuclein. Alpha-synuclein is known for its ability to aggregate. Could the authors comment on the suitability of replacing IRD2 with an IDP that is known to aggregate? Did the authors do additional checks with this mutant to check for aggregation? E.g. SEC/ultracentrifugation?
- Could the authors please include an SDS-PAGE of the purified CHD4 (and possibly also mutants) in the supplementary figures?
- In the single-molecule methods, a CHD4 binding assay is explained, but there is no such data in the paper.
- Figure 1B: All of the truncations used in the study are listed and visualised here, but none of the rearrangements (ie. The charge alterations to IDR1 and the synuclein sequence swap for IDR2)
- Regarding the above, given that much of the study involves assaying various mutants, the conclusion section would be aided with a table of all mutants and a brief summary of how they have been found to behave differently from the wild type.
- In Figure 6C (and 7B), a gamma distribution is fitted to the data, showing a decrease in Napp upon increasing ATP concentration. This is explained as evidence that the underlying rate distribution is not actually a gamma distribution and includes one additional step of a different rate to the other 5(?). This is a reasonable conclusion, but it may strengthen the findings of the study to build a model which incorporates one additional step (that is not ATP concentration-dependent) and then globally fit all data to this model. This could potentially recover the actual rate of each step and the number of ATP-dependent steps.
- Figure 7A: In both the examples for CHD4(1-1230) shown, the FRET drops to ~0 in what looks like a single step. Could this also be explained by photobleaching? In cases like this, where there is no visible pause time, how can you distinguish rapid sliding from photobleaching?
- Figure 7B: How do the authors determine a histogram for the pause time at mid-FRET for CHD4(1-1230) when the sliding is so fast that a pause at intermediate FRET is hard to make out (eg in the two examples in Fig 7A). It would be useful to mark the pause time on the FRET trajectories in Fig 7A in the same way that you indicated them in Fig 6B.

Reviewer #1:

Major:

1. In the “The N-terminal intrinsically disordered region regulates remodelling activity” section, what is the reason of “a N-terminal truncation that completely lacks IDR1 – CHD4(145–1912) – is even more active than WT (Figure 3B).”? Does this mean that the IDR1 is also an auto-inhibitory motif of the activity of CHD4? Furthermore, the activity comparison between IDR1 mutants and CHD4(355-1912) in the same section is not appropriate, since both the HMG and IDR2 except IDR1 were deleted in CHD4(355-1912).

This is a good point the reviewer raises. We do not currently have an explanation for this increase in activity, but we have added a sentence to the text to indicate the possibility that an element in the N-terminal region of CHD4 might play an auto-inhibitory role. Thus:

“Unexpectedly, an N-terminal truncation that completely lacks IDR1 – CHD4(145–1912) – is even slightly more active than WT (Figure 3B). It is thus possible that other elements in the very N-terminal region of CHD4 might inhibit CHD4 activity, but the mechanism underlying this difference is not currently clear.”

Regarding the comparison to CHD4(355-1912), we only made the comparison to try to make the point that the IDR1 mutants still retained considerable activity (whereas CHD4(355-1912) is hardly active at all). Given that this might be a confusing comparison, we have reworded the section to address the reviewer’s concern. Thus:

“In RTFA experiments, CHD4-IDR1-chrg1 and -chrg4 show slightly reduced activity compared to WT, -chrg2 and -chrg3 (Figure 3B), although all mutants retain considerable remodelling activity (Supplementary Figure 1).”

2. There are several comparisons of different truncations on nucleosome binding activity and remodeling behaviour in second paragraph on page 5, so what is the relationship between the substrate binding affinity and activity of CHD4? For most enzyme, the stronger substrate binding affinity, the higher activity. Is it fit for CHD4? If not, why?

This is an interesting point that the reviewer raises. We had also initially expected to see a clear correlation between substrate affinity and remodelling activity. However, a plot of dissociation constant for nucleosome binding (K_D) vs remodelling rate constant (k_{RTFA}) for all mutants (Figure R1) shows essentially no correlation. We propose that this lack of correlation arises because of the complex nature of the remodelling reaction – each cycle comprises multiple steps (multiple ATP hydrolysis events) and multiple cycles take place during the remodelling reaction that we measure in our fluorescence assays (because CHD4 is a processive enzyme).

Furthermore, the data we present in the manuscript show that there are multiple regulatory domains in the enzyme – both positive and negative in nature, and the set of mutants that we have studied cover the entire protein and therefore each affect a different aspect of the CHD4 remodelling mechanism. We therefore think that the combination of these different effects acts to obscure the relationship between substrate affinity and activity.

Figure R1. Plot of dissociation constant for nucleosome binding (K_D) vs remodelling rate constant (k_{RTFA}) for CHD4 mutants. The blue line is a linear regression curve, and the grey area is an estimated standard error for the fit at each point. It can be seen that the correlation is very poor ($R^2 < 0.1$).

3. The binding affinity to 46w60 DNA of C1bC2a [CHD4(1380–1800)] is significant weaker than the WT CHD4 to 46w60 DNA (10 μ M vs 350 nM), how can “the SANT-SLIDE domain acts to sense the presence of a nucleosomal substrate”?

Although the affinity of the isolated C1bC2a domain is significantly weaker than intact CHD4, our MST data show that it does contribute positively to substrate binding. **Supplementary Figure 1** shows that removal of C1bC2a reduces the DNA-binding affinity of CHD4 by ~ 3 -fold (compare 1–1380 with 1–1810). Even though this is a modest contribution, our hypothesis is that the interaction of this domain with, for example, linker DNA has the effect of displacing the adjacent auto-inhibitory C1a domain, thereby significantly increasing the remodelling activity of CHD4. We make this point in the Discussion section entitled “A DNA-binding SANT-SLIDE domain relieves auto-inhibition by the C1a region”.

Minor:

1. Figures numbering should be done according to the order of results, for example, figure 3D should swap with 3C, so 3C following 3B not 3D; figure 4A should also be cited in “Deletion of the C-terminal 102 residues..... three-fold (Figure 4B, Supplementary Figure 1).”; the MST data should be at the lowest position in supplementary figure 1, and so on.

These changes have been made.

2. Typo, it should be “remodeling” but not “remodelling”; “an N-terminal truncation” but not “a N-terminal truncation”; “the region C1a to C-terminal (residues 1340–1912)” but not “the region C-terminal to C1a (residues 1340–1912)”.

We have used “remodelling” because this was the spelling used in our previous paper that was published in Nature Communications. However, we are happy for this to be changed during the editing process if required. We have changed “a N-terminal truncation”. Regarding the last point, we think the reviewer has misunderstood this description. We are not describing the C1a region itself, but rather are referring to the region of CHD4 that is C-terminal to C1a. We have reworded this phrase to try to make it clearer:

“Our data show that the region of CHD4 that is C-terminal to C1a (i.e., residues 1340–1912) promotes...”

3. Carefully comparison indicated that the MST curve of WT CHD4 in figure 4B is same as that in figure 3E, so why there is difference of the scale or digital value of the y-axis?

Yes, they are the same curve. The y-axis are arbitrary units. The actual fluorescence values from different experiments can vary, and thus were scaled to the same starting value for easier comparison in **Figure 3E** – and we had meant to do the same for **Figure 4B**. We have now transformed **Figure 4B** in the same manner for consistency and have relabelled the y-axis as “Change in fluorescence (AU)”.

4. The label of the same information in different figures should be same, such as WT CHD4 in figures 3E and 4B, Napp and steps in figures 6C and 7B...

Yes – well spotted by the reviewer. We have made these edits.

5. What does the blank region mean in supplementary figure 1 of the MST binding data of IDR1 mutants? Furthermore, the unit of the KD should be nM but not μ M.

Blank regions indicate that no data were collected in those instances. We have added “ND” in these cases and added a definition to the figure legend. We have also changed the y-axis from μ M to nM.

6. The structural data need to be further refined, since the R values are so high and the gap between R_{work}/R_{free} is so big. Furthermore, there are many mistakes in structural table, such as the description of the unit cell, the format of CC1/2....

We have amended the text in the Materials and Methods to include more details on the structure quality. We agree with the reviewer that R_{free} is an important global measure of model-to-data agreement and as such we have tried to keep the R_{free} values as low as is acceptable at 2.9 Å resolution with flexible regions, without overfitting the data. Throughout the refinement procedure we have also tried to maintain a low gap between R_{work} and R_{free} achieving a final value of 4%. This is in line with the common rule of thumb to not exceed the gap by 5%, and we note that this value can potentially be much larger for low-resolution datasets¹⁻³.

We note that, in response to the reviewer’s comments, we undertook additional rounds of refinement. In so doing, we were able to reduce R_{free} by ~1%, but the stereochemical measures of structure quality were degraded, which is why we have retained the model reported in the original version of this manuscript.

We also thank the reviewers for pointing out the discrepancies in the refinement table. The errors have been rectified.

Reviewer #2 (Remarks to the Author):

Major Issues:

1. Figure 2A: the gel is of poor quality and makes it challenging to assess the authors' conclusions. Do the dashed lines indicate where gels have been cut and combined? Was the same amount loaded in each lane? By eye, it looks like the total amount in each lane varies. For lanes 2 and 3 (WT CHD4 and 1-1230), the gel shows roughly an equal split between slid and non-slid at equilibrium. However, in Fig 2C, the real-time fluorescence assays show that more nucleosomes slid by 1-1230 compared to WT. Therefore, a reader might expect to see more in the upper band for 1-1230 in the gel. Could the authors please redo the gel, either loading more or imaging for longer such that it is easier to see the bands? Using a fluorescently labelled nucleosome such as is used in Fig 6 and Fig 7 should make it easy to visualise the bands.

Yes, the gel lanes were combined from a larger gel and each lane was loaded with same quantity of nucleosome. We have added text to the figure legend to make these points clear. We have also rerun the

gel and the new version is now in **Figure 2A**. It can be challenging to compare the total band intensity for the starting nucleosome with that from lanes in which significant remodelling takes place, because the intensity in the latter case does not just arise from the two main bands but is also spread among several other less populated states.

Regarding the comparison between the fluorescence and gel-based assays, we do not have a clear explanation for why the apparent extent of remodelling differs for WT CHD4 vs CHD4(1–1230) in our gel based and fluorescence assays. It is worth noting that for WT CHD4 in gel based assays, we do not observe >50% of the nucleosome with a shifted retention time, even if we extend the incubation time or raise the concentration of ATP. We believe (but do not know for certain) that this represents some sort of equilibrium between the action of CHD4 and the higher thermodynamic stability of the end-positioned nucleosome (where the Widom sequence is found). It is also difficult to directly compare the extent of remodelling in the gel-based assays with that in the real-time fluorescence assay, because of differences in the nature of exactly what is being measured in each case. In the fluorescence assay, once the fluorophore is more than a certain distance from the quencher, we cease to see an increase in fluorescence, even if further remodelling takes place. In addition, the measured fluorescence is a weighted sum of the fluorescence of all species that are present at that point in time, whereas the gel-based assay resolves specific species (several shifted bands with intermediate retention times are often observed, depending on the construct and the reaction conditions). Overall, we avoid comparing the two assays in detail, but rather treat them as two different views of the remodelling reaction.

2. Figure 2C and D: In 2C, the max fluorescence reached is ~200 AU. In Figure 2D (which appears to be the fit of data from 2C), the Y axis shows a maximum of ~20 AU, while the bar itself is labelled with the number 2100. Could the authors please clarify these different values?

Also well spotted by the reviewer. The **Figure 2D** in the original manuscript is not the correct version - it was from an earlier draft. We have now gone through and standardized the fluorescence changes (as noted above) so that all of the Figures are consistent.

We have also changed the y-axis label in all RTFA plots to “Change in fluorescence (AU)” for consistency.

3. Figure 2C: the figure caption states: “t is the time in minutes after addition of CHD4 and ATP”. Could the authors clarify if CHD4 and ATP were added simultaneously? From Fig 2C, the dashed line indicates that only ATP was injected. The materials and methods section also has conflicting information on whether CHD4 and nucleosome were preincubated together before the addition of ATP or if ATP and CHD4 were added at the same time.

CHD4 was mixed with nucleosome before the start of experiment and ATP was added 5 min later. The variable t is thus the time in minutes after addition of CHD4. We have edited the legend to **Figure 2C** to include this information.

4. Figure 2D: the figure caption says that the rate and max fluorescence values were determined by fitting to a logarithm function. Why was a logarithm function chosen here and an exponential function in Fig 2C?

This is a typographic error. All RTFA data were fitted to the asymptotic exponential equation described in the legend of **Figure 2C**. We have corrected the legend for **Figure 2D** accordingly.

5. Figure 3C right panel: The fluorescence increase for both the WT and del- mutant appear to be biphasic in this experiment, with a fast phase and a slow phase. Elsewhere the WT fluorescence increase looks to be single exponential. Could the authors comment on this, please?

We carried out three replicates for this measurement, and the data shown in **Figure 3C** of the manuscript are the mean values from the three replicates. We agree that it looks biphasic, but have no clear explanation for this observation. Below, we show the three individual replicates, together with fits for these replicates to the same model, as well as the average data – also fitted to this model. Some variation in shape is evident between the traces, but all overall yield similar values and reasonable fits to the model. We agree that it is possible that there may be an additional process at play here, but it would be an entire

project of its own to try to get to the bottom of what that might be. We have noted the shape of this curve in the legend to **Figure 3**. We note here that – by showing the mean data rather than the ‘best’ looking individual replicate – we at least show the ‘warts’ to the reader so that they can see that the behaviour of that mutant is not ‘perfect’. We hope this is satisfactory.

Figure R2. Individual RTFA curves for replicates of the del- mutant.

6. In the text, when describing the K_d values determined by MST, it would also be useful to state the error in each value.

We have included an estimate of uncertainty in the text (p4) as well as in all MST figures, in line with the 25% that we state in the legend to **Figure 3**.

7. Figure 4A: The data presented here for the 1-1230 truncation looks like it could be the same data as is presented in Fig 2C for the same truncation. Is this the exact same data, and are the authors including it again for comparison? If so, it should be stated clearly in the caption that this data is duplicated from a different figure for the purpose of ease of comparison. Could the authors please confirm that no other data has been presented in multiple panels without explicitly stating such?

Yes, these are the same data, which have been reproduced for comparison as indicated by the reviewer. We have amended the legend of **Figure 4** to reflect this reproduction (and likewise for MST data for WT CHD4 in **Figure 4B**, which were also shown in **Figure 3E**).

8. Figures 6 and 7: For the example FRET trajectories given, it is hard to judge the FRET value without any additional y-axis ticks. In the methods section, it is stated that molecules displaying a FRET value between 0.55 and 0.7 were included in the analysis (proximal labelled molecules). Judging by eye, many of the example trajectories shown (particularly in Fig 7A) start at a FRET value that looks >0.7 . Are the authors confident that these high FRET molecules are proximal labelled and not double labelled molecules?

The reviewer is correct in that a number of the traces shown have starting FRET values >0.7 . These were shown by mistake – we had selected individual traces for display in the Figure from the full set of traces we obtained in each experiment, and a lack of attention to detail resulted in the selection of a number of traces that have a higher probability of being doubly labelled. We have replaced these traces with ones that are more likely to be single labelled. We emphasize though that only traces with starting FRET values of 0.55–0.7 were used in the numerical analysis, and so none of our conclusions are altered in any way.

Minor Issues:

1. Page 2, paragraph 3: it may read better if the word “formed” was replaced with “form” in the sentence beginning, “The translocase domain is immediately preceded by...”

This text has been changed as suggested.

2. For the CHD4-IRD2-del+ mutant, the IRD2 region is replaced with (almost the entire length of) alpha-synuclein. Alpha-synuclein is known for its ability to aggregate. Could the authors comment on the suitability of replacing IRD2 with an IDP that is known to aggregate? Did the authors do additional checks with this mutant to check for aggregation? E.g. SEC/ultracentrifugation?

We clarified the protein by high-speed centrifugation (20,000xg for 30 min) and confirmed that the protein was still in supernatant, before proceeding to experiments. Further, the stock protein concentration was kept low at all times (<100 nM), whereas the dissociation constant for the formation of alpha-synuclein dimers (a first step in the aggregation process) has been reported to be ~ 2 mM (<https://www.biorxiv.org/content/10.1101/795997v1.full.pdf>), meaning that the chance of IRD2 forming alpha-synuclein-like aggregates is low, particularly as it is flanked by considerable additional protein sequence.

3. Could the authors please include an SDS-PAGE of the purified CHD4 (and possibly also mutants) in the supplementary figures?

An SDS-PAGE of purified CHD4 has been added as **Supplementary Figure 7**.

4. In the single-molecule methods, a CHD4 binding assay is explained, but there is no such data in the paper.

This text was carried over from a previous manuscript. It has been removed.

5. Figure 1B: All of the truncations used in the study are listed and visualised here, but none of the rearrangements (ie. The charge alterations to IDR1 and the synuclein sequence swap for IDR2)

We have provided the sequences for the rearrangement constructs in **Supplementary Figure 2**. In line with the reviewer’s suggestion, we have edited **Figure 1** to indicate the positions of these mutations.

6. Regarding the above, given that much of the study involves assaying various mutants, the conclusion section would be aided with a table of all mutants and a brief summary of how they have been found to behave differently from the wild type.

Supplementary Figure 1 summarizes the behaviour of all mutants used in the study. It includes both the affinity of each mutant for the nucleosome substrate (from MST) and the rate and extent of remodelling as measured in the RTFA assays.

7. In Figure 6C (and 7B), a gamma distribution is fitted to the data, showing a decrease in Napp upon increasing ATP concentration. This is explained as evidence that the underlying rate distribution is not actually a gamma distribution and includes one additional step of a different rate to the other 5(?). This is a reasonable conclusion, but it may strengthen the findings of the study to build a model which incorporates

one additional step (that is not ATP concentration-dependent) and then globally fit all data to this model. This could potentially recover the actual rate of each step and the number of ATP-dependent steps.

Our simulations, shown in **Supplementary Figure 5**, show that the observed decrease of the number of kinetic intermediates N_{app} with increasing ATP concentration is consistent with the existence of an additional process which is not dependent on ATP hydrolysis. However, the simulation also shows that the existence of an additional non-ATP dependent step would lead to a pause time distribution undistinguishable from a gamma-distribution with the observed N_{app} and k_{app} . In fact, neither N_{app} nor k_{app} are a monotonic function of k_N/k_x and two different ratios k_N/k_x can give rise to a mathematically identical distribution. Fitting to such a model would therefore be by nature non-reproducible. Because of this complication, we prefer to report confident fits to a simplified model, rather than ambiguous/unfaithful fits to what might nevertheless be a more accurate model.

8. Figure 7A: In both the examples for CHD4(1-1230) shown, the FRET drops to ~ 0 in what looks like a single step. Could this also be explained by photobleaching? In cases like this, where there is no visible pause time, how can you distinguish rapid sliding from photobleaching?

Photobleaching of the acceptor would result in FRET values that are *exactly* zero [because $FRET = I_{acc}/(I_{acc} + I_{don})$] and thus traces displaying photobleaching are easily distinguishable from traces that transition to low FRET. We excluded from our analysis all traces where either donor or acceptor dye bleaches during a relevant transition.

9. Figure 7B: How do the authors determine a histogram for the pause time at mid-FRET for CHD4(1-1230) when the sliding is so fast that a pause at intermediate FRET is hard to make out (eg in the two examples in Fig 7A). It would be useful to mark the pause time on the FRET trajectories in Fig 7A in the same way that you indicated them in Fig 6B.

Pause times were determined by an unbiased change-point algorithm. Trajectories where only one change in FRET value was detected were excluded from the histograms. The observed rate is therefore a conservative estimate.

References

- 1 Shabalin, I. G., Porebski, P. J. & Minor, W. Refining the macromolecular model - achieving the best agreement with the data from X-ray diffraction experiment. *Crystallogr Rev* **24**, 236-262, doi:10.1080/0889311X.2018.1521805 (2018).
- 2 Wlodawer, A. Stereochemistry and Validation of Macromolecular Structures. *Methods Mol Biol* **1607**, 595-610, doi:10.1007/978-1-4939-7000-1_24 (2017).
- 3 Wlodawer, A., Minor, W., Dauter, Z. & Jaskolski, M. Protein crystallography for non-crystallographers, or how to get the best (but not more) from published macromolecular structures. *FEBS J* **275**, 1-21, doi:10.1111/j.1742-4658.2007.06178.x (2008).

REVIEWERS' COMMENTS

Reviewer #1 (Remarks to the Author):

This revision addresses all of my concerns in full, so I recommend for publication.

Reviewer #2 (Remarks to the Author):

The authors have satisfactorily addressed my concerns. The revised ms should be of interest to the broad readership of Nat Commun.